# Seg2Any: Open-set Segmentation-Mask-to-Image Generation with Precise Shape and Semantic Control

**Danfeng Li**[1,2]* **Hui Zhang**[1,2]* **Sheng Wang**[3] **Jiacheng Li**[3] **Zuxuan Wu**[1,2]†

[1]Shanghai Key Lab of Intell. Info. Processing, School of CS, Fudan University
[2]Shanghai Collaborative Innovation Center of Intelligent Visual Computing
[3]HiThink Research

https://seg2any.github.io

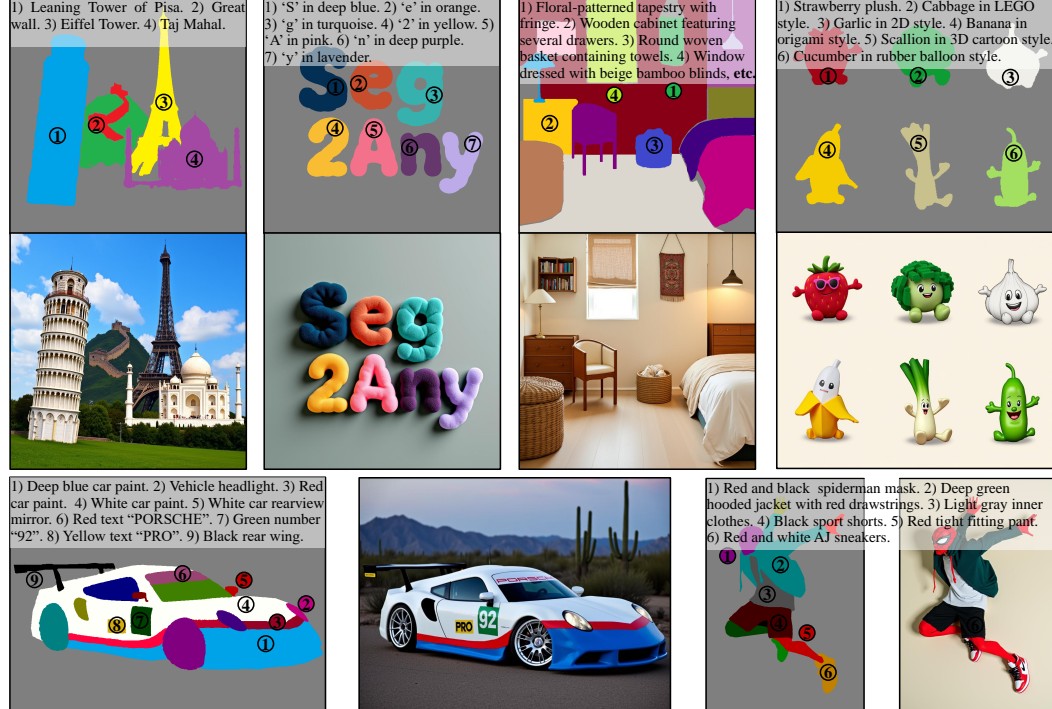

Figure 1: We propose Seg2Any, a novel segmentation-mask-to-image generation approach that achieves strong shape consistency and fine-grained attribute control (*e.g.* color, style, and text).

## Abstract

Despite recent advances in diffusion models, top-tier text-to-image (T2I) models still struggle to achieve precise spatial layout control, *i.e.* accurately generating entities with specified attributes and locations. Segmentation-mask-to-image (S2I) generation has emerged as a promising solution by incorporating pixel-level spatial guidance and regional text prompts. However, existing S2I methods fail to simultaneously ensure semantic consistency and shape consistency. To address these challenges, we propose Seg2Any, a novel S2I framework built upon advanced multimodal diffusion transformers (*e.g.* FLUX). First, to achieve both semantic

---

*Equal contribution.

†Corresponding author.

39th Conference on Neural Information Processing Systems (NeurIPS 2025).

and shape consistency, we decouple segmentation mask conditions into regional semantic and high-frequency shape components. The regional semantic condition is introduced by a Semantic Alignment Attention Mask, ensuring that generated entities adhere to their assigned text prompts. The high-frequency shape condition, representing entity boundaries, is encoded as an Entity Contour Map and then introduced as an additional modality via multi-modal attention to guide image spatial structure. Second, to prevent attribute leakage across entities in multi-entity scenarios, we introduce an Attribute Isolation Attention Mask mechanism, which constrains each entity's image tokens to attend exclusively to themselves during image self-attention. To support open-set S2I generation, we construct SACap-1M, a large-scale dataset containing 1 million images with 5.9 million segmented entities and detailed regional captions, along with a SACap-Eval benchmark for comprehensive S2I evaluation. Extensive experiments demonstrate that Seg2Any achieves state-of-the-art performance on both open-set and closed-set S2I benchmarks, particularly in fine-grained spatial and attribute control of entities.

# 1   Introduction

Text-to-image (T2I) generation [34, 4, 9] has been widely adopted in various applications due to its powerful generative capabilities. However, T2I models struggle to achieve precise layout control (*i.e.* precisely generate entities in specified attributes and positions) solely through text prompts.

Layout-to-image generation has been proposed and designed to generate images based on specified layout conditions, including spatial locations and descriptions of entities. These layout conditions come in various forms, such as bounding boxes [22, 60, 44, 19, 54, 25], depth maps [62, 63, 61], segmentation masks [47, 24, 53], *etc*. This paper focuses on segmentation-mask-to-image (S2I) generation, where segmentation masks dictate the spatial locations of entities, and text descriptions specify their semantic content, thereby enabling the most fine-grained control over the images.

Existing S2I methods can be mainly divided into two categories: I) Methods that integrate segmentation masks as additional conditional inputs, such as ControlNet [56], ControlNet++ [20] and T2I-Adapter [26]. These methods often fail to align regional textual descriptions with their respective regions, resulting in semantic inconsistency in the generated images (see Figure 2 (a)). II) Methods based on the masked attention mechanism, which restricts each text embedding to attend solely to the respective image embeddings (*e.g.* FreestyleNet [47], PLACE [24] and EliGen [53]). Although these methods achieve semantic alignment, they fall short in precise shape preservation, as shown in Figure 2 (b) and (c). We attribute this shape inconsistency to the loss of spatial information when segmentation masks are compressed into the latent space. Notably, compared to the UNet [34] architecture with $8\times$ downsampling, recent advanced DiT [28] architectures employ a more aggressive $16\times$ downsampling, which further amplifies the loss of spatial information.

To address these challenges, we propose Seg2Any, a novel S2I framework built upon advanced multimodal diffusion transformers (*e.g.* FLUX [18]). Seg2Any mainly relies on two key innovations: I) **Semantic-Shape Decoupled Layout Conditions Injection**. We decouple segmentation mask conditions into two components: the shape condition (high-frequency shape information) and the regional semantic condition (low-frequency semantic information). *For the injection of regional semantic conditions, we employ a Semantic Alignment Attention Mask.* This mechanism tightly binds each entity's image tokens to its corresponding text prompts, ensuring that the generated image is semantically consistent with the input descriptions at the entity level. *For the injection of shape condition, we propose Sparse Shape Feature Adaptation to integrate key spatial structure into the model efficiently.* Conventional approaches [56, 20] assign fixed colors to distinct categories in the semantic segmentation maps, which is only suitable for closed-set S2I generation and cannot be generalized to open-set scenarios. In contrast, we introduce an Entity Contour Map as our category-agnostic shape representation, which consists of entity contours extracted from segmentation masks. This representation is inherently sparse, preserving essential shape details in a compact and efficient manner. Following OminiControl [37], we integrate the condition tokens with text and noisy image tokens into a unified sequence, allowing them to interact directly through multi-modal attention [9]. As shown in Figure 2 (d), our approach achieves both semantic and shape consistency simultaneously.

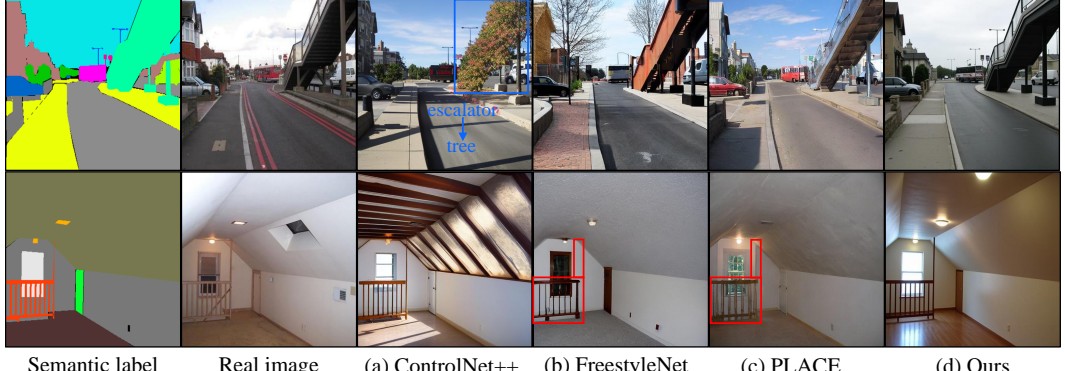

| Semantic label | Real image | (a) ControlNet++ | (b) FreestyleNet | (c) PLACE | (d) Ours |

Figure 2: Comparison in terms of shape and semantic consistency. Semantic inconsistency is annotated by blue boxes, while shape inconsistency is highlighted with red boxes, which reveal inconsistency in the number of vertical bars on the railings. In contrast, our approach achieves both shape and semantic consistency.

II) **Attribute Isolation via Image Self-Attention Mask**. Multi-instance generation often suffers from attribute leakage, where the visual attributes of one entity may affect others. To mitigate this problem, we introduce an Attribute Isolation Attention Mask strategy that prevents cross-entity information leakage by ensuring that image tokens associated with each entity are isolated from those of others.

Current dense caption datasets [33, 51, 31, 8, 50] are often limited by their closed-set vocabularies and coarse-grained descriptions, which constrain their effectiveness in training S2I models. Recent advances in open-source vision language models (VLMs), such as Qwen2-VL-72B [43], have significantly reduced the performance gap with close-source VLMs like GPT-4V [1], making it feasible to create large-scale and richly annotated datasets. Leveraging the capabilities of Qwen2-VL-72B, we construct Segment Anything with Captions 1 Million (SACap-1M), a large-scale dataset derived from the diverse and high-resolution SA-1B dataset [16]. SACap-1M contains 1 million image-text pairs and 5.9 million segmented entities, each comprised of a segmentation mask and a detailed regional caption, with captions averaging 58.6 words per image and 14.1 words per entity. We further present the SACap-Eval, a benchmark for assessing the quality of open-set S2I generation.

Through comprehensive evaluations on both open-set (SACap-Eval) and closed-set (COCO-Stuff, ADE20K) benchmarks, Seg2Any consistently outperforms prior SOTA models, particularly in fine-grained spatial and attribute control of entities.

To summarize, our contributions are as follows:

1. We propose Seg2Any, a framework that enables precise control over shape and semantics while preventing attribute leakage in open-set S2I generation.

2. We construct SACap-1M, a large-scale open-set dataset with 1M images and 5.9M regional annotations, along with SACap-Eval, an open-set benchmark for evaluating S2I generation.

3. Seg2Any achieves state-of-the-art performance on both open-set (SACap-Eval) and closed-set (COCO-Stuff, ADE20K) benchmarks.

## 2  Related Work

### 2.1  Text-to-Image Generation

Text-to-image (T2I) generation [34, 4, 9, 42, 38] has undergone significant advancement in recent years. Motivated by advances in large-scale transformer architectures, the Diffusion Transformer (DiT) [28] was introduced. Building upon this foundation, recent models like SD3 [9] and FLUX [18] further propose the Multimodal Diffusion Transformer (MM-DiT), which treats text as an independent modality and incorporates flow matching objectives, achieving state-of-the-art results. To integrate additional condition images (e.g., canny maps, depth maps, subject references) into MM-DiT, OminiControl [37] introduces a novel controllable framework. It concatenates condition image, text, and noisy image tokens, and then employs task-specific LoRA [13] modules to handle various conditions within a unified pipeline while maintaining minimal trainable parameters.

## 2.2 Layout-to-Image Generation

Layout-to-Image (L2I) generation is a task that synthesizes images guided by spatial layout conditions and entity-level textual descriptions. Existing L2I methods can be categorized according to the type of layout condition they employ. These include: **Bounding box-based approaches** [22, 60, 44, 19, 54, 55, 25, 11, 10, 36, 7, 58, 48] typically use rectangular spatial constraints to guide the image generation process, offering a coarse-grained layout control. **Depth map-based approaches** [62, 63, 61] utilize depth maps to achieve fine-grained spatial control similar to segmentation masks. For instance, 3DIS [62] divides multi-instance generation into two stages: it first generates a depth map via a text-to-depth model, then uses a pre-trained depth-to-image model to synthesize images with multi-instance attribute control. DreamRenderer [61], a training-free approach based on pre-trained controllable T2I models (FLUX.1-Depth [18] and FLUX.1-Canny [18]), identifies that middle layers in the FLUX model are responsible for instance-level rendering while shallow and deep layers capture global context. Consequently, it applies a hard image self-attention mask only to the middle layers to prevent attribute leakage. However, as a training-free approach, directly applying the attention mask severely impairs overall visual harmony.

**Segmentation mask-based approaches** [47, 24, 53, 27, 15, 2, 52] focus on pixel-level layout control. For example, FreestyleNet [47] uses binary attention weights in the cross-attention module that assigns a value 1 to allow text tokens to bind to corresponding image regions and 0 to prevent attention from unrelated areas. However, this approach requires downsampling the segmentation masks to align with the lower-resolution latent features, thereby sacrificing spatial detail. To alleviate the above issue, PLACE [24] introduces a layout control map that softens the attention weights. Instead of hard binary assignments in FreestyleNet, it calculates the area proportions of different entities within the receptive field of every image token, yielding soft attention weights. Yet, this approach struggles to mitigate spatial detail loss in MM-DiT architectures, where 16× downsampling (compared to 8× in U-Net) leads to more severe degradation of spatial information. The work most similar to ours is EliGen [53], which is also built on FLUX. Unlike our approach, it is trained on datasets with bbox annotations and only supports loose position control through scribble-style masks, whereas our approach enables both strict and loose mask position control. Additionally, it relies solely on a masked attention mechanism without injecting explicit spatial information.

## 3 Methodology

### 3.1 Problem Definition

We define the instruction $y$ for a segmentation-mask-to-image model as a composition of a global text prompt and $N$ entity-level text prompts with corresponding binary masks:

$$y = [p_0, (p_1, m_1), \ldots, (p_i, m_i), \ldots, (p_N, m_N)], i \in [1, N], \quad (1)$$

where $p_0$ denotes the global textual description, while each $p_i$ represents the entity-specific textual prompt with its corresponding binary segmentation mask $m_i$ for the $i$-th entity.

### 3.2 Semantic-Shape Decoupled Layout Conditions Injection

As shown in Figure 3, we decouple segmentation mask-based layout conditions into complementary semantic and shape components. For semantic information, we employ a **Semantic Alignment Attention Mask** (Section 3.2.1) mechanism that binds text prompts to their corresponding image regions. For shape information, we adopt the **Sparse Shape Feature Adaptation** (Section 3.2.2) to efficiently encode and integrate spatial layout conditions into the model.

#### 3.2.1 Semantic Alignment Attention Mask

To ensure semantic alignment between textual and visual modalities, we introduce a **Semantic Alignment Attention Mask**, denoted as $\mathbf{M}_{\text{sem-align}}$, which governs the interactions between textual and visual tokens. We denote the text token indices of the global caption as $\mathcal{T}_0$, and those of the $i$-th entity's regional caption as $\mathcal{T}_i$ ($i = 1, \ldots, N$). Likewise, $\mathcal{I}_0$ and $\mathcal{I}_i$ correspond to the image token indices of the background and the $i$-th entity, respectively. The $\mathbf{M}_{\text{sem-align}}$ is then defined as follows:

$$\mathbf{M}_{\text{sem-align}}[q, k] = \begin{cases} 1, & \text{if } q \in \mathcal{T}_i, \ k \in \mathcal{T}_i \quad \text{(text-text)} \\ 1, & \text{if } q \in \mathcal{T}_i \cup \mathcal{T}_0, \ k \in \mathcal{I}_i \quad \text{(text-image)} \\ 1, & \text{if } q \in \mathcal{I}_i, \ k \in \mathcal{T}_i \cup \mathcal{T}_0 \quad \text{(image-text)} \quad , i \in [0, N], \\ 1, & \text{if } q, k \in \bigcup_{i=0}^{N} \mathcal{I}_i \quad \text{(image-image)} \\ 0, & \text{otherwise} \end{cases} \quad (2)$$

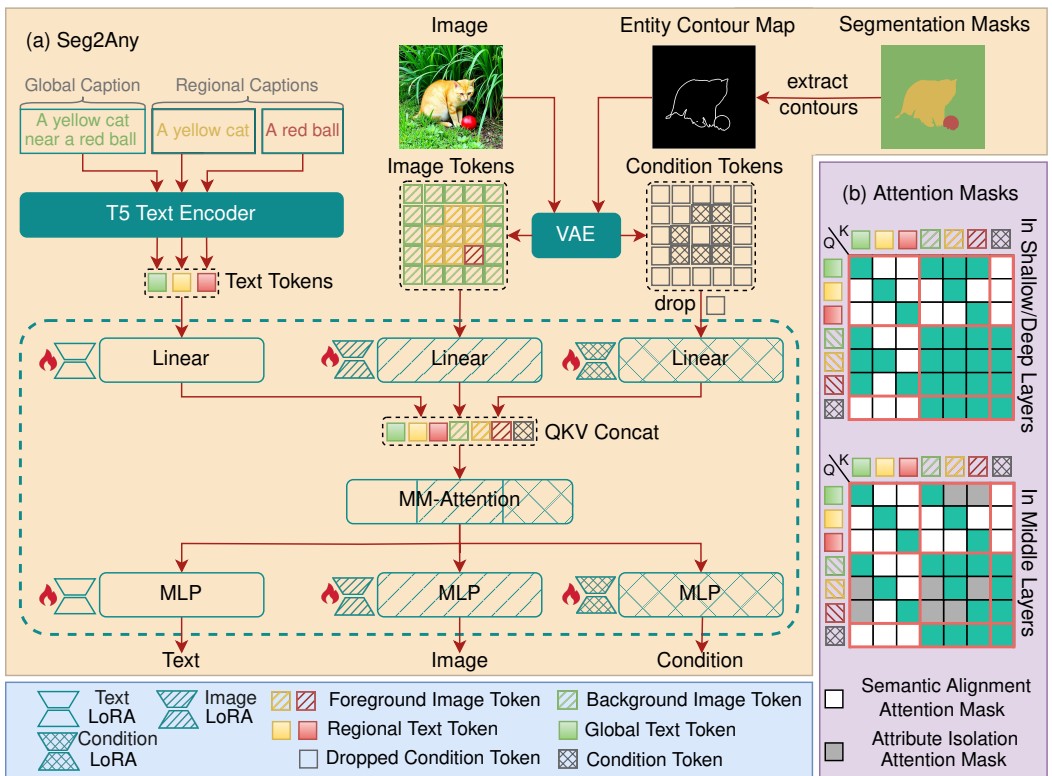

Figure 3: (a) An overview of the Seg2Any framework. Segmentation masks are transformed into Entity Contour Map, then encoded as condition tokens via frozen VAE. Negligible tokens are filtered out for efficiency. The resulting text, image, and condition tokens are concatenated into a unified sequence for MM-Attention. Our framework applies LoRA to all branches, achieving S2I generation with minimal extra parameters. (b) Attention Masks in MM-Attention, including Semantic Alignment Attention Mask (Section 3.2.1) and Attribute Isolation Attention Mask (Section 3.3).

where $q$ and $k$ are the query and key indices, respectively. As illustrated in Figure 3 (b), this attention mechanism guarantees that each generated entity adheres closely to its text prompt. Meanwhile, all image tokens attend to each other to ensure globally coherent visual synthesis. The global caption tokens $\mathcal{T}_0$ are allowed to attend to all image tokens, providing contextual global guidance.

### 3.2.2 Sparse Shape Feature Adaptation

**Condition Image Representation.** In open-set S2I generation, the common practice of using fixed, class-specific colors to represent semantic segmentation maps is inherently limited. To address this limitation, we use an **Entity Contour Map** to effectively encode shape information.

Starting with a set of binary masks $\{m_i\}_{i=1}^N$ for $N$ distinct entities where each $m_i \in \{0,1\}^{H \times W}$, we extract the contour of each mask as $\text{Contour}(m_i) \in \{0,1\}^{H \times W}$. These contours are then merged into a single binary map:

$$C_{\text{gray}}(x,y) = \max_{1 \le i \le N} \text{Contour}(m_i)(x,y). \tag{3}$$

The resulting grayscale map $C_{\text{gray}} \in \mathbb{R}^{H \times W}$ is then further converted to an RGB image by duplicating the gray channel across all three channels, resulting in our Entity Contour Map $C \in \mathbb{R}^{H \times W \times 3}$.

This sparse shape representation offers advantages in our framework, as semantic information is already integrated through the Semantic Alignment Attention Mask mechanism (Section 3.2.1). This approach eliminates the need for dense, per-pixel semantic labels to indicate region occupancy. Due to its efficiency and sparsity, we adopt this representation as the shape encoding method in our work.

**Minimal Condition Image Control.** Inspired by OminiControl [37], we treat the Entity Contour Map—a type of image-based condition—as an independent modality and leverage LoRA [13] to

minimize training overhead. As shown in Figure 3 (a), the Entity Contour Map is encoded into condition image tokens by a frozen VAE encoder, concatenated with text and noisy image tokens to form the joint input sequence, which directly participates in multi-modal attention. Furthermore, the condition tokens share the same 2D position indices with the noisy image tokens under the RoPE encoding, which helps preserve spatial alignment.

Notably, unlike OminiControl, which applies LoRA only to the condition branch. We apply LoRA to all branches (as shown in Figure 3 (a)). The condition branch is trained with LoRA to seamlessly incorporate the Entity Contour Map. Meanwhile, the image and text branches are also trained using LoRA to ensure precise alignment between the generated entities and the regional text prompts. This approach modifies the linear layers in each DiT block across all three branches as follows:

$$
\begin{aligned}
W_{\text{cond}}^{\text{new}} &= W_{\text{img}} + B_{\text{cond}} A_{\text{cond}}, \\
W_{\text{img}}^{\text{new}} &= W_{\text{img}} + B_{\text{img}} A_{\text{img}}, \\
W_{\text{text}}^{\text{new}} &= W_{\text{text}} + B_{\text{text}} A_{\text{text}},
\end{aligned}
\tag{4}
$$

where $A_{\text{cond}}, A_{\text{img}}, A_{\text{text}} \in \mathbb{R}^{r \times d}$ and $B_{\text{cond}}, B_{\text{img}}, B_{\text{text}} \in \mathbb{R}^{d \times r}$ are the low-rank adaptation matrices for each respective branch ($r \ll d$). Here, $W_{\text{img}}, W_{\text{text}} \in \mathbb{R}^{d \times d}$ denote the original weight matrices of the linear layers.

**Shape Guidance Strength Modulation.** To adjust the influence of condition image tokens, the attention mechanism is adapted by incorporating a bias term [37], defined as:

$$
\text{Attention}(Q, K, V) = \text{softmax}\left( \frac{QK^\top}{\sqrt{d}} + \text{Bias}(\gamma) \right) V,
\tag{5}
$$

where $Q, K, V$ are computed from the concatenation of text tokens, noisy image tokens, and condition tokens. The bias matrix $\text{Bias}(\gamma) \in \mathbb{R}^{(L_{\text{text}} + 2L_{\text{img}}) \times (L_{\text{text}} + 2L_{\text{img}})}$, with $L_{\text{text}}$ and $L_{\text{img}}$ denoting the number of text and image-related tokens respectively, is given by:

$$
\text{Bias}(\gamma) = \begin{bmatrix} 0_{L_{\text{text}} \times L_{\text{text}}} & 0_{L_{\text{text}} \times L_{\text{img}}} & 0_{L_{\text{text}} \times L_{\text{img}}} \\ 0_{L_{\text{img}} \times L_{\text{text}}} & 0_{L_{\text{img}} \times L_{\text{img}}} & \log(\gamma) \cdot \mathbf{1}_{L_{\text{img}} \times L_{\text{img}}} \\ 0_{L_{\text{img}} \times L_{\text{text}}} & \log(\gamma) \cdot \mathbf{1}_{L_{\text{img}} \times L_{\text{img}}} & 0_{L_{\text{img}} \times L_{\text{img}}} \end{bmatrix}.
\tag{6}
$$

The factor $\gamma \in (0, 1]$ serves to modulate the strength of shape conditioning. As $\gamma$ approaches 1, $\log(\gamma)$ tends to 0, and the condition tokens retain full influence, enforcing strict adherence to the shape guidance. In contrast, as $\gamma$ approaches 0, $\log(\gamma)$ tends to $-\infty$, which suppresses the contribution of condition tokens, thus making scribble-style segment masks feasible for flexible control.

**Condition Image Token Filtering.** Given the sparse nature of the Entity Contour Map, where numerous areas exhibit low or zero values, we discard tokens that are entirely composed of zero values, as they provide no shape information. This results in a significant reduction in tokens without compromising shape details. The token filtering process is illustrated in Figure 3 (a).

### 3.3 Attribute Isolation Attention Mask

A critical challenge in multi-instance generation is attribute leakage, where visual attributes from one entity transfer to others (as illustrated in Figure 4). To address this, we introduce the **Attribute Isolation Attention Mask**, defined as $\mathbf{M}_{\text{attr-isolate}}$:

$$
\mathbf{M}_{\text{attr-isolate}}[q, k] = \begin{cases} 1, & \text{if } q \in \mathcal{T}_i, \ k \in \mathcal{T}_i \quad \text{(text-text)} \\ 1, & \text{if } q \in \mathcal{T}_i, \ k \in \mathcal{I}_i \quad \text{(text-image)} \\ 1, & \text{if } q \in \mathcal{I}_i, \ k \in \mathcal{T}_i \quad \text{(image-text)} \\ 1, & \text{if } q \in \mathcal{I}_i \cup \mathcal{I}_0, \ k \in \mathcal{I}_i \quad \text{(image-image)} \\ 0, & \text{otherwise} \end{cases} \quad , i \in [0, N],
\tag{7}
$$

Unlike the Semantic Alignment Mask (Eq. 2), the Attribute Isolation Mask operates with stricter constraints, as illustrated in Figure 3 (b). First, it prevents cross-entity visual information interaction by restricting each entity's image tokens to attend only to themselves. Second, it restricts the global caption tokens ($\mathcal{T}_0$) from attending to any foreground image tokens. This complete separation of entities effectively prevents attribute leakage. Notably, background image tokens are still permitted to attend to all image tokens, ensuring environmental coherence.

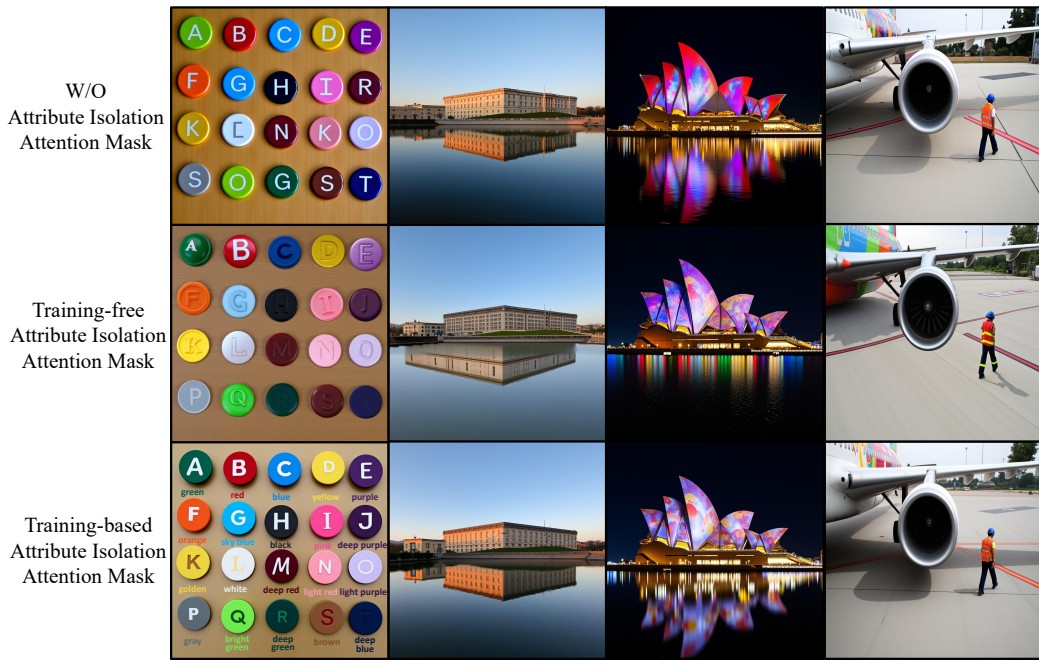

Figure 4: Visualization results of different attribute isolation strategies. In Column 1, 20 colored circular badges labeled A to T are required to be generated in raster order. The results show that our Attribute Isolation Attention Mask effectively prevents attribute leakage between entities. Columns 2-4 demonstrate that direct application of the mask without training leads to visual inconsistencies, manifesting as unnatural shadows and reflections. In contrast, the training-based approach on our proposed large-scale dataset achieves both strong attribute control and high visual coherence.

Building on insights from DreamRender [61] that the middle layers (20-38 layers) of the 57-layer FLUX architecture are dedicated to processing the visual features of individual instances, we apply the Attribute Isolation Attention Mask to these middle layers. In practice, we observe that training-free rigid attention constraints often introduce visual artifacts, as demonstrated in Figure 4. Instead, through training on our proposed large-scale datasets, deeper layers learn to refine holistic image quality, achieving an optimal balance between visual harmony and attribute control.

## 3.4 SACap-1M Dataset

To address the lack of large-scale and fine-grained datasets for S2I generation, we construct SACap-1M, containing 1 million image-text pairs and 5.9 million segmented entities with detailed descriptions. We propose an automated pipeline for data annotation and filtering: I) **Image Filtering.** We select images from the high-resolution and wide-ranging SA-1B [16] dataset. Initially, we remove images that are excessively large or small in size, as well as those with extreme aspect ratios. Subsequently, we apply the LAION-Aesthetics predictor [35] to filter out low-quality images with an aesthetic score below 5. II) **Entity Extraction.** The SA-1B dataset provides accurate, category-agnostic masks for each image. However, each image contains on average over 100 masks, many of which are nested. To filter masks, we retain only top-level masks by removing those that are contained within top-level masks. Additionally, we discard masks whose area is smaller than $1\%$ of the total image area. Finally, we exclude images whose number of remaining masks falls outside the range of 1 to 20. III) **Regional and Global Caption Annotation.** We employ the open-source Qwen2-VL-72B [43] model to generate captions for each entity and the global image, yielding an average of 58.6 words per image and 14.1 words per entity. See supplemental materials for more details.

# 4 Experiments

## 4.1 Experiment Setup

**Baselines.** Our method is compared with prior state-of-the-art S2I approaches, including FreestyleNet [47] and PLACE [24]. Controllable T2I models, such as ControlNet [56], Control-Net++ [20], and UniControl [29], are also considered. Additionally, recent FLUX-based models (DreamRender [61], 3DIS [63], and EliGen [53]) are included in the comparison. Notably, Dream-Render and 3DIS are training-free methods that require depth maps as input instead of segmentation masks. For evaluation of these models, we use Depth Anything V2 [49] to predict depth maps from real images as guidance.

**Training datasets and Evaluation Benchmarks.** Experiments are conducted on both open-set and closed-set segmentation datasets. *For the open-set scenario*, we utilize SACap-1M, which consists of 1 million images accompanied by 5.9 million regional captions. Evaluation for this setting is performed on SACap-Eval, a benchmark curated from a subset of SACap-1M, comprising 4,000 prompts with detailed entity descriptions and corresponding segmentation masks, with an average of 5.7 entities per image. *For closed-set scenario*, we select two widely used datasets: ADE20K [59] and COCO-Stuff [3]. Following prior works [47, 24], for ADE20K and COCO-Stuff, regional captions are assigned as the semantic class names of each segment, and no global caption is provided.

**Implementation Details.** Our experiments are conducted based on FLUX.1-dev. The LoRA modules are applied to all linear layers of each block in DiT, with the LoRA rank set to 64, resulting in approximately 594M additional parameters. Across all datasets, our model is trained for 20,000 steps with a batch size of 16, using the AdamW optimizer and a fixed learning rate of 0.0001. The training resolution is set to $1024 \times 1024$ for the SACap-1M dataset and $512 \times 512$ for ADE20K and COCO-Stuff. All experiments are conducted on 4 NVIDIA H100 GPUs.

**Evaluation Metrics.** *For closed-set S2I generation*, we report both FID and MIoU. FID reflects the visual fidelity of the generated images, while MIoU measures semantic and layout consistency. For MIoU calculation, we use Mask2Former [6] for ADE20K and DeepLabV3 [5] for COCO-Stuff to predict semantic segmentation, as done in ControlNet++ [20]. *For open-set S2I generation*, following CreatiLayout [54], we evaluate S2I quality on SACap-Eval from three perspectives:

- **Class-agnostic MIoU.** We use the ground-truth segmentation masks as mask prompts, and then employ SAM2 [32] to predict class-agnostic segmentation masks for the generated images. The predicted masks are compared with the ground truth to compute the class-agnostic MIoU, which measures shape consistency.

- **Region-wise quality.** Image regions are cropped based on ground-truth segmentation masks and evaluated with Qwen2-VL-72B [43] in a Visual Question Answering (VQA) manner to measure both spatial and attribute accuracy. Specifically, spatial accuracy is evaluated by checking whether each entity appears within the correct region, while attribute accuracy considers whether its color, text, and shape match the provided descriptions.

- **Global-wise quality.** We assess overall visual quality and global caption fidelity using multiple metrics, including the scoring models IR score [46] and Pick score [17], as well as commonly used metrics such as CLIP score and FID.

## 4.2 Quantitative Results

**Fine-Grained open-set S2I.** Table 1 presents the quantitative results on the SACap-Eval benchmark. We report multiple metrics covering class-agnostic shape consistency, region-wise, and global-wise qualities. Compared to previous methods, Seg2Any achieves significant improvements across most criteria. Notably, our approach reaches a class-agnostic MIoU of 94.90, nearly matching the real image upper bound (96.03 class-agnostic MIoU), which demonstrates excellent shape consistency. In terms of region-wise assessment, our method excels in region-wise spatial localization and precise attribute control, effectively mitigating attribute leakage. The qualitative results presented in Figure 5 further support this observation. For global quality, our method achieves strong overall performance thanks to the FLUX model and high-quality training data.

Table 1: Quantitative comparison on the SACap-Eval benchmark. **Bold** and underline represent the best and second best methods, respectively.

| Method | Class-agnostic MIoU ↑ | Region-wise Quality | | | | Global-wise Quality | | | |
|---|---|---|---|---|---|---|---|---|---|
| | | Spatial ↑ | Color ↑ | Shape ↑ | Texture ↑ | IR [46] ↑ | Pick [17] ↑ | CLIP ↑ | FID ↓ |
| Real Images | 96.03 | 97.04 | 93.87 | 91.66 | 92.50 | - | - | - | - |
| FreestyleNet [47] | 74.59 | 42.34 | 40.08 | 25.07 | 40.07 | -1.96 | 18.13 | 19.66 | 46.20 |
| PLACE [24] | 84.30 | 79.05 | 49.40 | 52.00 | 57.96 | -1.11 | 19.71 | 24.69 | 17.81 |
| EliGen [53] | 51.38 | 83.62 | 76.94 | 78.91 | 77.21 | 0.49 | **22.60** | 27.57 | 19.69 |
| 3DIS [63] | 82.09 | 88.25 | 85.80 | 85.87 | 89.20 | **0.53** | 21.81 | 28.03 | 15.36 |
| DreamRender [61] | 82.32 | 88.69 | 87.73 | 86.76 | 89.19 | 0.43 | 21.58 | **28.21** | **13.71** |
| **Seg2Any (ours)** | **94.90** | **93.89** | **91.52** | **88.15** | **90.12** | 0.44 | 21.66 | 27.87 | 15.53 |

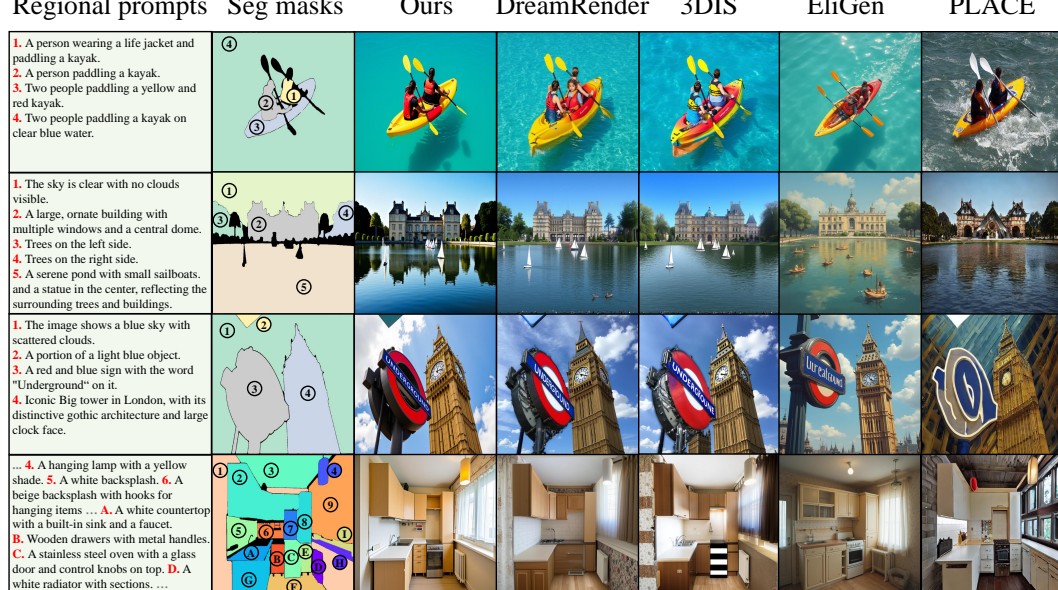

Figure 5: Qualitative comparisons on SACap-Eval. Seg2Any accurately generates entities exhibiting complex attributes such as color and texture, surpassing previous approaches.

**Coarse-Grained Closed-set S2I.** Table 2 shows quantitative results for closed-set S2I generation on the COCO-Stuff and ADE20K datasets. On COCO-Stuff, Seg2Any achieves the highest MIoU. On ADE20K, Seg2Any reaches 54.46 MIoU, approaching the upper bound of real images (54.41 MIoU). Although Seg2Any does not surpass PLACE (60.20 MIoU) on ADE20K, it outperforms all other baselines, demonstrating strong overall performance across datasets. Seg2Any exhibits higher FID scores compared to PLACE and FreestyleNet, which we attribute to the domain gap between the base model FLUX and the real images in ADE20K and COCO-Stuff.

## 4.3 Ablation Study

We conduct an ablation study to evaluate the individual contributions of our key components on the SACap-Eval benchmark (see Table 4). Introducing explicit shape condition via Sparse Shape Feature Adaptation (SSFA) significantly improves the class-agnostic MIoU metric over using only the Semantic Alignment Attention Mask (SAA), confirming the effectiveness of preserving shape details. Further, adding the Attribute Isolation Attention Mask (AIA) notably improves region-wise quality, indicating better prevention of attribute leakage. The training-free version degrades global quality; however, after training on our large-scale dataset, the overall image quality improves significantly. Table 4 also shows that incorporating Condition Image Token Filtering (CITF) leaves performance almost unchanged; its impact on computational cost is further analyzed in the supplemental materials.

On COCO-Stuff and ADE20K, we observe a similar trend: SSFA substantially improves MIoU, and CITF shows a marginal impact on performance. (see Table 3).

Table 2: Quantitative evaluation of S2I models on COCO-Stuff and ADE20K.

| Method | COCO-Stuff | | ADE20K | |
|---|---|---|---|---|
| | MIoU ↑ | FID ↓ | MIoU ↑ | FID ↓ |
| Real Images | 40.87 | - | 54.41 | - |
| FreestyleNet [47] | 42.42 | 15.12 | 44.42 | 28.45 |
| PLACE [24] | 42.23 | **14.95** | **60.20** | **24.51** |
| GLIGEN [22] | - | - | 23.78 | 33.02 |
| ControlNet [56] | 27.46 | 21.33 | 32.55 | 33.28 |
| UniControl [29] | - | - | 25.44 | 46.34 |
| Controlnet++ [20] | 34.56 | 19.29 | 43.64 | 29.49 |
| **Seg2Any (ours)** | **45.54** | 19.90 | 54.46 | 32.89 |

Table 3: Ablation results on COCO-Stuff and ADE20K. We conduct ablation experiments on SAA (Semantic Alignment Attention Mask), SSFA (Sparse Shape Feature Adaptation), and CITF (Condition Image Token Filtering).

| SAA | SSFA | CITF | COCO-Stuff | | ADE20K | |
|---|---|---|---|---|---|---|
| | | | MIoU ↑ | FID ↓ | MIoU ↑ | FID ↓ |
| ✓ | - | - | 43.48 | 20.57 | 44.85 | 33.14 |
| ✓ | ✓ | - | 45.50 | **19.06** | 54.11 | **32.37** |
| ✓ | ✓ | ✓ | **45.54** | 19.90 | **54.46** | 32.89 |

Table 4: Ablation results on the SACap-Eval benchmark. MIoU* denotes class-agnostic MIoU. The ablation focuses on SAA (Semantic Alignment Attention Mask), SSFA (Sparse Shape Feature Adaptation), AIA† (Training-Free Attribute Isolation Attention Mask) AIA‡ (Training-Based Attribute Isolation Attention Mask), and CITF (Condition Image Token Filtering).

| Methods | | | | | MIoU* | Region-wise Quality | | | | Global-wise Quality | | | |
|---|---|---|---|---|---|---|---|---|---|---|---|---|---|
| SAA | SSFA | AIA† | AIA‡ | CITF | | Spatial ↑ | Color ↑ | Shape ↑ | Texture ↑ | IR ↑ | Pick ↑ | CLIP ↑ | FID ↓ |
| ✓ | - | - | - | - | 89.39 | 93.36 | 89.26 | 85.84 | 87.86 | **0.47** | **21.83** | **28.09** | 15.30 |
| ✓ | ✓ | - | - | ✓ | 94.16 | 93.43 | 89.16 | 85.61 | 88.53 | 0.44 | 21.81 | 27.98 | **15.20** |
| ✓ | ✓ | ✓ | - | ✓ | 94.46 | 92.37 | **92.16** | 87.77 | **90.23** | 0.27 | 21.39 | 27.79 | 16.11 |
| ✓ | ✓ | - | ✓ | - | **94.94** | 93.36 | 90.08 | 87.40 | 88.93 | 0.39 | 21.57 | 27.97 | 17.35 |
| ✓ | ✓ | - | ✓ | ✓ | 94.90 | **93.89** | 91.52 | **88.15** | 90.12 | 0.44 | 21.66 | 27.87 | 15.53 |

## 5 Conclusion

We propose Seg2Any, a novel segmentation-mask-to-image generation framework that achieves fine-grained layout control by decoupling spatial layout and semantic guidance. Through the integration of sparse entity contours and multi-modal masked attention, Seg2Any simultaneously ensures shape preservation, semantic alignment, and robust attribute control. We further introduce the large-scale SACap-1M dataset and SACap-Eval benchmark to foster open-set S2I research. Extensive experiments validate that Seg2Any achieves new state-of-the-art performance, particularly excelling in generating entities with detailed descriptions.

**Limitation.** Seg2Any faces resource constraints when generating images with a large number of entities, each accompanied by detailed descriptions. Additionally, our large-scale dataset relies on vision-language models for regional captioning, which inevitably introduces annotation noise that may impact segmentation-mask-to-image generation performance.

## 6 Acknowledgment

This work was supported in part by the National Natural Science Foundation of China (Grant 62472098).

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

# A    More Details of Condition Image Token Filtering

We analyze the computational cost of Condition Image Token Filtering (CITF) on three benchmark datasets: COCO-Stuff, ADE20K, and SACap-1M. For each dataset, we consider five settings to comprehensively analyze the computational cost: I) without condition tokens, representing the lowest computation. II–IV) CITF applied with the minimum, maximum, and average numbers of retained condition tokens, respectively. V) without CITF, i.e., no token filtering, representing the highest computation. For each configuration, we report the resulting average image generation time and multiply–accumulate operations (MACs).

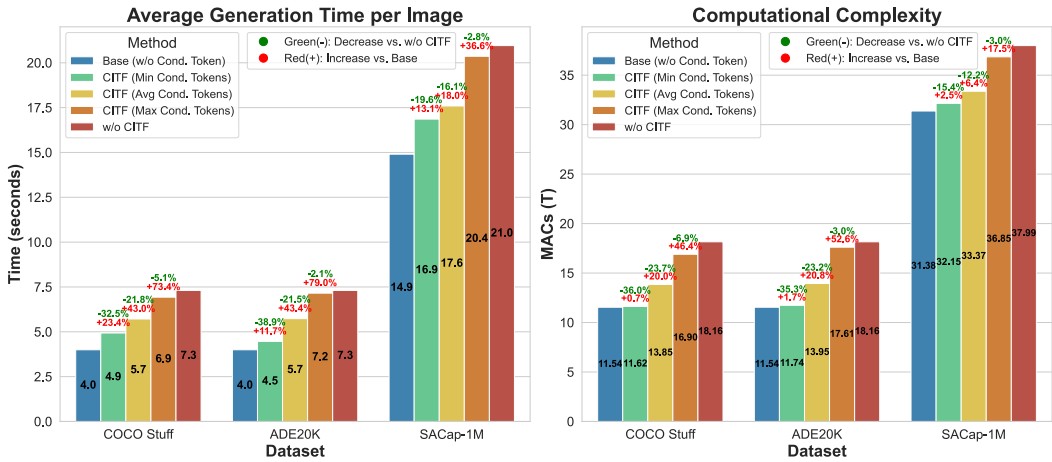

Figure 6: Impact of Condition Image Token Filtering (CITF) on computational cost.

Figure 6 illustrates the results. All experiments are conducted on a single NVIDIA H100 GPU with 32 sampling steps. Each image uses 5 regional text prompts (50 tokens each) and one global prompt (512 tokens). As shown, CITF leads to a notable reduction in both inference time and MACs when the entity contour maps are sparse. When more tokens are retained, computational savings decrease. Overall, CITF provides a simple yet effective mechanism to reduce inference overhead, particularly in cases with sparse entity layouts.

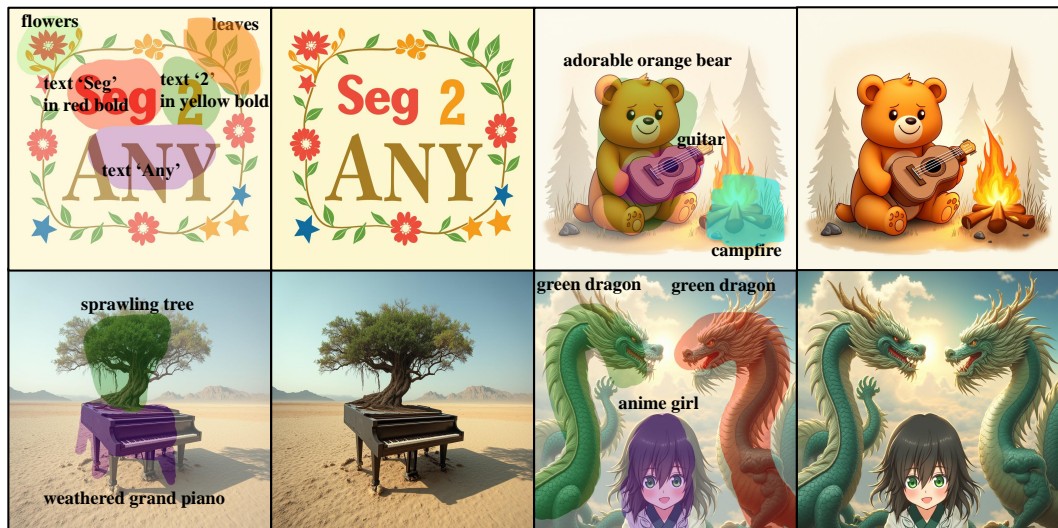

Figure 7: Qualitative results with shape guidance strength ($\gamma = 0.2$), demonstrating flexible scribble-style control.

# B  More Details of Shape Guidance Strength Modulation

We empirically find that adjusting the strength hyperparameter $\gamma$ in Shape Guidance Strength Modulation serves as a "free lunch", enabling scribble-style control without incurring any additional training overhead. In our experiments, we set $\gamma = 0.2$, which provides sufficient flexibility for scribble-style segment masks while still maintaining semantic alignment. The qualitative results are shown in Figure 7.

# C  More Ablation Study Results

## C.1  Ablation on AIA

To further verify the effectiveness of the proposed Attribute Isolation Attention Mask (AIA), we conduct an ablation experiment by removing the AIA module from Seg2Any. As illustrated in Figure 8, the absence of AIA leads to attribute leakage across entities.

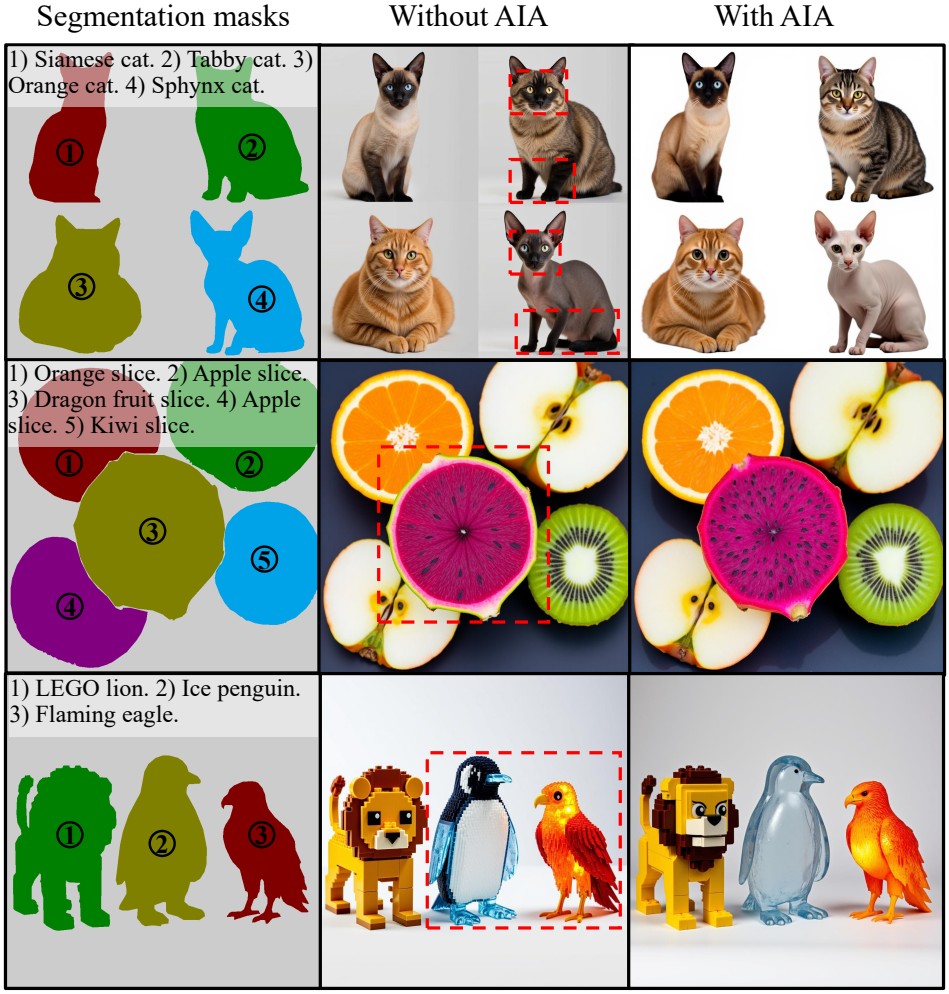

Figure 8: More qualitative ablation comparisons on the Attribute Isolation Attention Mask (AIA). Without AIA, attribute leakage occurs across entities, e.g., fur texture is transferred between cat breeds (row 1), the red peel of dragon fruit is replaced by kiwi's green peel (row 2), and the LEGO style spreads to other entities (row 3). In contrast, with AIA, such attribute leakage is effectively eliminated.

## C.2 Ablation on SSFA vs. PLACE Attention

To ensure a fair comparison between the proposed Sparse Shape Feature Adaptation (SSFA) and PLACE attention [24], we reimplemented PLACE attention [24] under the same FLUX architecture, while keeping all other training setups fixed. Specifically, we use a learning rate of 1e-4, a batch size of 16, and train for 20k steps on 4 NVIDIA H100 GPUs. As shown in Tables 5 and 6, our model surpasses this PLACE attention variant by (+2.55%) MIoU on ADE20K and (+3.36%) class-agnostic MIoU on SACap-Eval, confirming the superiority of our design in a strictly fair comparison.

Table 5: Comparison with PLACE attention under the FLUX model on COCO-Stuff and ADE20K.

| Method | COCO-Stuff | | ADE20K | |
|---|---|---|---|---|
| | MIoU ↑ | FID ↓ | MIoU ↑ | FID ↓ |
| SAA | 43.48 | 20.57 | 44.85 | 33.14 |
| PLACE attention | 44.56 | 20.10 | 51.56 | **32.22** |
| SAA+SSFA | **45.50** | **19.06** | **54.11** | 32.37 |

Table 6: Comparison with PLACE attention under the FLUX model on SACap-Eval.

| Method | Class-agnostic MIoU ↑ | Region-wise Quality | | | | Global-wise Quality | | | |
|---|---|---|---|---|---|---|---|---|---|
| | | Spatial ↑ | Color ↑ | Shape ↑ | Texture ↑ | IR ↑ | Pick ↑ | CLIP ↑ | FID ↓ |
| SAA | 89.39 | 93.36 | **89.26** | 85.84 | 87.86 | 0.47 | 21.83 | **28.09** | 15.30 |
| PLACE attention | 90.8 | 93.03 | 89.12 | **86.36** | 88.08 | **0.51** | **21.90** | 27.93 | 15.81 |
| SAA+SSFA | **94.16** | **93.43** | 89.16 | 85.61 | **88.53** | 0.44 | 21.81 | 27.98 | **15.20** |

Figure 9 presents the synthesis results on the ADE20K dataset using SAA, PLACE attention, and SAA+SSFA, all trained under the same FLUX architecture. It can be seen that SAA+SSFA generates shapes that best conform to the provided segmentation masks.

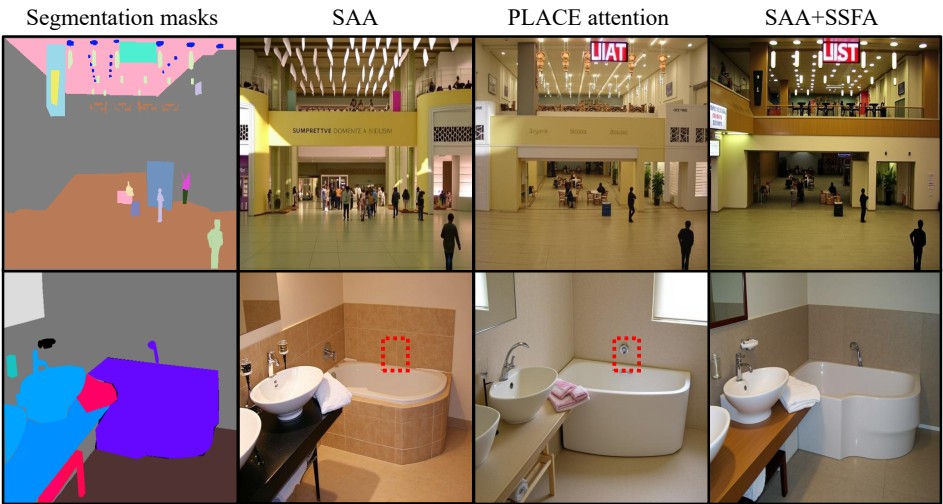

Figure 9: More qualitative ablation comparisons on the Sparse Shape Feature Adaptation (SSFA). All methods are trained under the same FLUX architecture.

# D    More Qualitative Results

We provide additional qualitative results in Figure 10 to illustrate the effectiveness of our approach. Seg2Any shows strong capability in generating high-quality images that faithfully adhere to detailed entity descriptions, enabling flexible and precise control over complex visual attributes such as color, texture, and shape. Figure 11 presents the qualitative results of Seg2Any in densely crowded scenes.

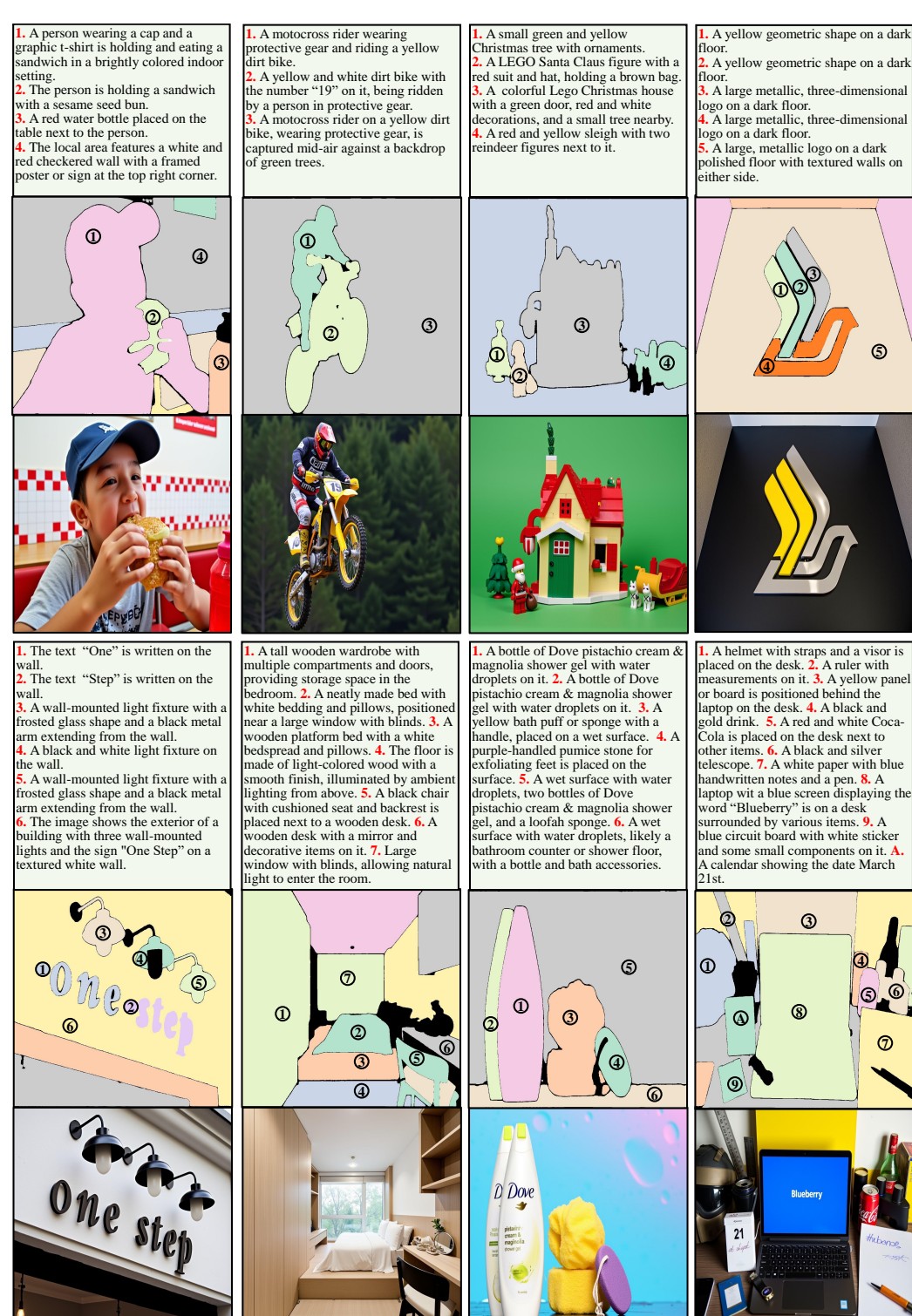

Figure 10: More qualitative results using the Seg2Any method.

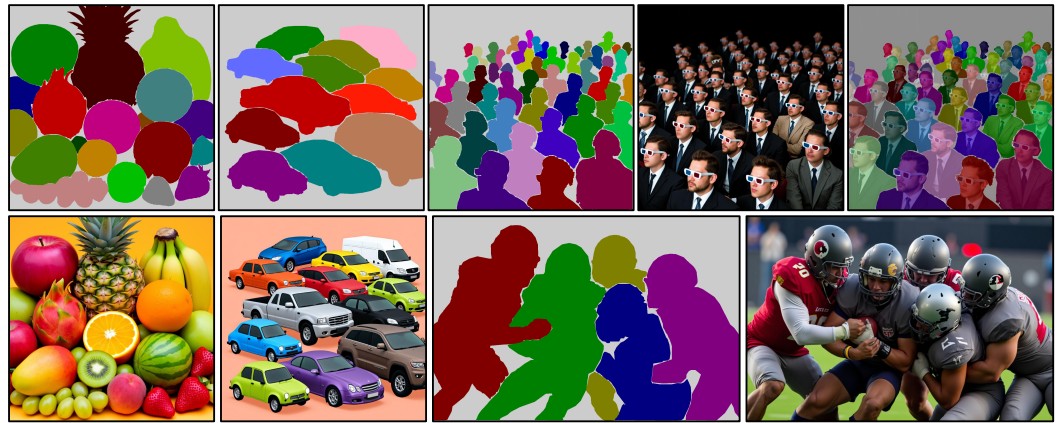

Figure 11: Qualitative results of Seg2Any in densely crowded scenes.

# E  More Details on Datasets and Benchmarks

## E.1  Comparison with Existing Dense Caption Datasets

Table 7 presents a detailed comparison between our SACap-1M dataset and several recent dense caption datasets. Unlike previous datasets such as PixelLM-MUSE [33], Osprey [51], COCONut-PanCap [8] and Pix2Cap-COCO [50] which are constructed upon closed-set label spaces (e.g., LVIS [12] and COCO [23]), our proposed SACap-1M dataset provides open-set segmented entities, enabling much broader generalization and flexibility in open-set segmentation-mask-to-image generation. Compared to GLaMM-GranD [31], also derived from SA-1B [16], our SACap-1M achieves significantly higher segmentation mask density and caption granularity by employing the advanced open-source Qwen2-VL-72B [43] vision-language model to generate more accurate and fine-grained regional captions.

Table 7: Comparison of dense caption datasets. Note that "Avg. Words" indicates the word count per regional caption, and "Avg. Masks" denotes the average number of masks per image.

| Dataset Name | Image Source | Image Number | Annotated by | Avg. Words | Avg. Masks |
|---|---|---|---|---|---|
| PixelLM-MUSE [33] | LVIS [12] | 246K | GPT-4V [1] | 11.3 | 3.7 |
| Osprey [51] | COCO [23] PACO-LVIS [30] | 724K | GPT-4V [1] | 38.7 | - |
| GLaMM-GranD [31] | SA-1B [16] | 11M | GPT4RoI [57] GRiT [45] | 5.8 | 4.4 |
| COCONut-PanCap [8] | COCO [23] | 118K | GPT-4V [1] and human correction | 16.6 | 13.2 |
| Pix2Cap-COCO [50] | COCO [23] | 20K | GPT-4V [1] and human correction | 22.94 | 8.14 |
| SACap-1M (ours) | SA-1B [16] | 1M | Qwen2-VL-72B [43] | 14.1 | 5.9 |

## E.2  Data annotation pipeline

We construct our dataset from the SA-1B [16] dataset through a multi-stage filtering and annotation process:

I) **Image Filtering.** Images are filtered based on size and aspect ratio, keeping those with width and height between 1000 and 3000 pixels and aspect ratios from 0.6 to 1.8. To further ensure visual quality, images with a LAION-Aesthetics [35] score below 5 are excluded.

II) **Entity Extraction.** While SA-1B provides numerous accurate, class-agnostic segmentation masks—averaging over 100 per image—many are nested and redundant. To mitigate this redundancy,

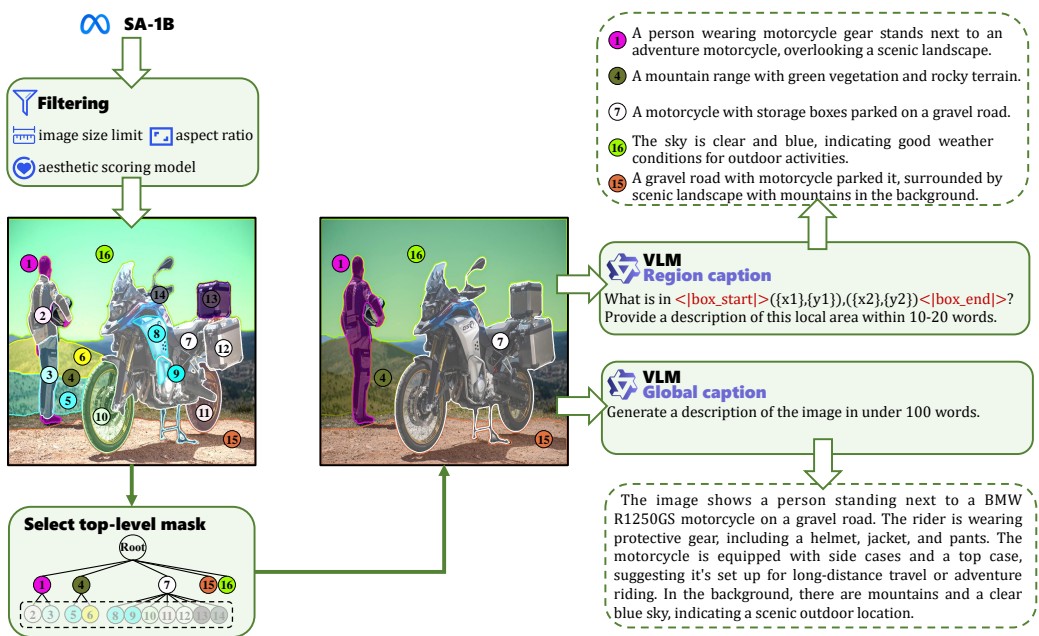

Figure 12: An overview of the data annotation pipeline.

we retain only the top-level masks, which generally correspond to the primary objects in the image, and discard any masks fully enclosed by others (as shown in Figure 12). Additionally, masks occupying less than 1% of the image area are considered too small and removed. Finally, only images containing between 1 and 20 valid masks are retained for further annotation.

III) **Regional and Global Caption Annotation.** We utilize the vision-language model Qwen2-VL-72B [43] for both regional and global captioning. For regional captioning, we incorporate the bounding box coordinates of each segment into the text prompt (see Figure 12), enabling the model to produce accurate and context-aware descriptions of each local region. This approach yields detailed annotations, with an average of 14.1 words per entity and 58.6 words per image, supporting fine-grained segmentation-to-image generation.

Finally, we present SACap-1M, a large-scale layout dataset containing 1 million image-text pairs and approximately 5.9 million segmented entities. As illustrated in Figure 13, most images contain a moderate number of masks, while images with a large number of masks constitute the long tail of the distribution. Figure 14 shows some examples from SACap-1M.

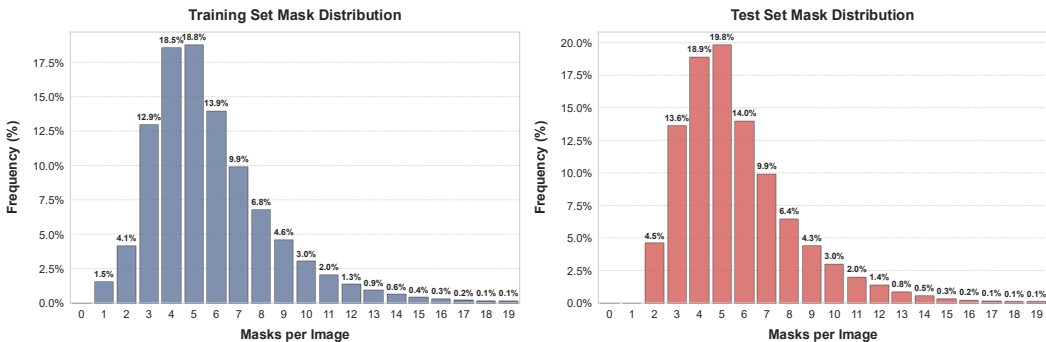

Figure 13: The distribution of the number of segmentation masks per image across the training and test sets.

1. A pink perfume that reads "My Scent" and has a small emblem on it.
2. A string of pearls and beads on a beige surface.
3. A green leaf is partially visible on the right side of the image.
4. Two pink triangular earrings with purple beads attached to them.
5. A set of pink and white pearl-like beads arranged on a surface.

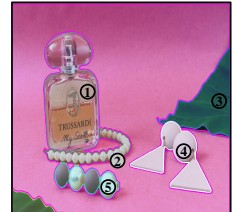

1. A shirtless man wearing white boxing gloves and black shorts with white trim.
2. A pair of black boxing shorts with white trim and text.
3. A shirtless boxer in black shorts with white trim, preparing to throw a punch.
4. A man's short black hair.

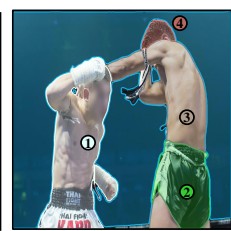

1. A white wooden block calendar displaying the date "December 25".
2. A small decorated Christmas tree with red ornaments and a red bow on top.
3. A decorative cloth or ribbon wrapped around the base of a small Christmas tree.
4. A small, decorated Christmas tree with red ornaments and a burlap base.
5. A cardboard robot figure standing next to a small decorated Christmas tree.
6. A wooden surface.

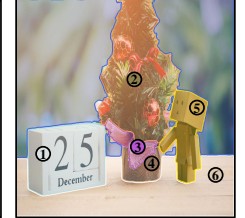

1. A person wearing a helmet and protective gear riding a yellow motorcycle on a paved road.
2. The front wheel of a yellow motorcycle on a paved road.
3. A motorcyclist wearing protective gear and a helmet is riding on an asphalt road with green grass on the side.
4. A grassy area with some yellow flowers located next to a paved road.
5. Grassy area next to the road.

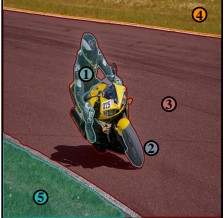

1. A bottle of Pepsi with a blue cap is placed on the table. 2. A black disposable cup with a white lid. 3. A slice of bread with a dark spread on it. 4. A white plate with a piece of bread covered in a dark spread. 5. A tray with food including bread, potatoes, vegetables, and meat, along with drink and utensils. 6. A slice of bread is placed on the tray next to the plate of food. 7. A plate with potatoes vegetables, and meat on a tray. 8. A wooden table with a tray containing food, a drink, and utensils, set outdoors on a stone patio. 9. A stone-paved walkway is visible behind the table and bench.

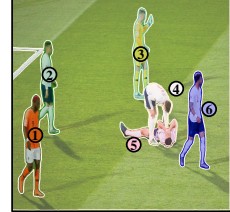

1. A calm body of water with several boats and buoys visible in the distance. 2. A woven wooden structure. 3. A person wearing a blue shirt with pink and white stripes, carrying a backpack standing near the water. 4. A backpack black straps . 5. A person's face with headphones on looking forwards at the water. 6. A pair of black headphones with a cushioned headband, and ear cups. 7. A person with short dark hair wearing headphones and sunglasses. 8. The sky is clear and blue with no visible clouds.

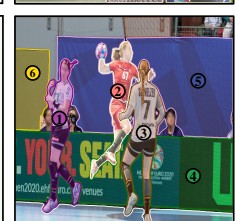

1. Lush green tree with dense foliage stands near the water's edge. 2. A tree with green leaves stands near the building. 3. A traditional Chinese-style building with an orange roof and red pillars, situated near a body of water. 4. A white boat-like structure with intricate designs, floating on the water near the building. 5. There are trees behind the building. 6. A traditional Chinese-style building with an orange roof and red pillars, reflecting in the water. 7. A calm body of water reflecting the surrounding structures and trees. 8. The sky is blue with some white clouds scattered throughout.

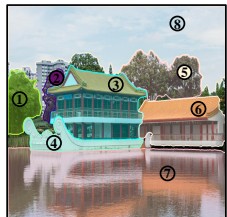

1. A handball player in a white jersey and black shorts, wearing white shoes with blue accents, preparing to receive the ball. 2. A handball player in a red jersey with the number 67, jumping and holding a ball. 3. A handball player in a white jersey with the number 7, jumping and attempting to block the ball. 4. A blue and green scoreboard with text and numbers displayed on it. 5. A blue wall with white text and logos, likely part of a sports facility or arena. 6. A gray wall with a rectangular window or panel.

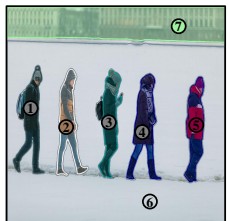

1. A soccer player wearing an orange jersey and white shorts stands on the field.
2. A soccer player wearing a white jersey and blue shorts, standing on the field.
3. A soccer player wearing yellow uniform standing on the field.
4. A soccer player in white and blue uniform standing on the field.
5. A soccer player lying on the ground with his hands covering his face.
6. A soccer player wearing a white jersey and blue shorts is standing on the field.

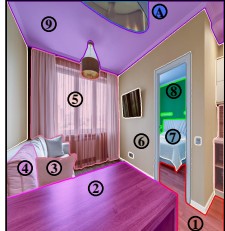

1. A person wearing a black coat and hat, carrying a backpack, walking on a snowy path. 2. A person wearing a brown and black jacket, blue jeans, and carrying a backpack, walking on snow. 3. A person wearing a black coat scarf, and hat walking on snow. 4. A person wearing a brown coat and blue jeans walking on snow. 5. A person wearing a red jacket and hat walking on snow. 6. A snowy landscape with five people walking across it, wearing winter clothing and carrying backpacks. 7. A row of buildings with multiple windows, likely residential or commercial structures, set against a snowy backdrop.

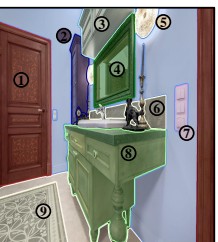

1. A wooden floor with light-colored wood planks and a smooth finish. 2. A wooden table or desk with a smooth surface and light brown color. 3. A beige throw pillow on the couch. 4. A white pillow. 5. Large windows with sheer curtains and wooden blinds, allowing natural light to enter the room. 6. A cozy living room with a wooden table, white sofa, and large windows. A blue bedroom is visible through an open door. 7. A blue bedspread on the bed. 8. A blue wall with a white shelf and a doorway leading to another room. 9. The ceiling features a modern light fixture and a decorative cornice along the edges. A. A ceiling light fixture with a warm amber glow.

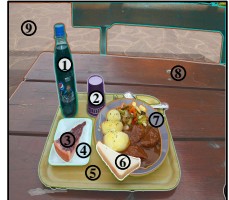

1. A wooden door with decorative panels and a doorknob. 2. A wooden cabinet with glass doors. 3. A wooden cabinet with a mirror above it and a light fixture on the wall. 4. A large mirror with an ornate frame is mounted above the vanity. 5. A decorative light fixture with a textured glass shade hangs from the ceiling. 6. A decorative backsplash with intricate patterns and a statue of a horse on top. 7. A white light switch panel with multiple switches is mounted on the wall. 8. A wooden vanity with a black countertop, a sink, and a decorative mirror above it. 9. A decorative floor tile with a geometric pattern featuring swirls and lines.

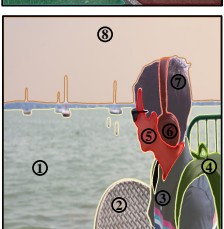

Figure 14: Examples from the SACap-1M dataset.

### E.3 SACap-Eval Benchmark

We construct SACap-Eval, a benchmark derived from SACap-1M, designed to assess the quality of segmentation-mask-to-image generation. The benchmark comprises 4,000 samples, with an average of 5.7 entities per image. Evaluation is conducted from two perspectives: Spatial and Attribute. Both aspects are assessed using the vision-language model Qwen2-VL-72B [43] via a visual question answering manner.

**Spatial Score.** For each segmentation mask, we first crop the corresponding region from the image and then prompt the VLM to determine whether the target entity is located within this area, allowing responses of either "Yes" or "No". The spatial score is obtained by computing the ratio of "Yes" answers to the total number of entities.

**Attribute Score.** To compute the attribute score, the region specified by each segmentation mask is cropped from the image, after which the VLM determines whether the entity inside this area satisfies the described attributes. Each attribute (*e.g.* color, shape, or texture) is examined separately using visual question answering, and the score is calculated in the same manner as the spatial score.

## F   Broader Impacts

The proposed method for segmentation-mask-to-image synthesis has potential applications in controllable image generation [21, 14], video generation [40, 41, 39], and the construction of image segmentation datasets. However, like other generative models, its misuse could result in the production of misleading or inappropriate content. Our approach may also inherit biases present in the training datasets, potentially reinforcing certain stereotypes. Responsible usage and further investigation into fairness and robustness are important, but a comprehensive analysis is beyond the scope of this work.

