# OpenReview forum: "Seg2Any: Open-set Segmentation-Mask-to-Image Generation with Precise Shape and Semantic Control"
_NeurIPS.cc/2025/Conference — NeurIPS 2025 poster_

### Official Review · Reviewer_Q4gs · 2025-06-16

**Clarity:** 2
**Significance:** 2
**Originality:** 2
**Rating:** 3
**Confidence:** 4

**Summary:**

This paper proposes a segmentation mask-to-image framework that enables precise control over shape and semantics while preventing attribute leakage in open-set S2I generation, including the injection of semantic-shape decoupled layout conditions and attribute isolation via an image self-attention mask. A large-scale open-set dataset, along with a corresponding benchmark, is proposed.

**Questions:**

1. I am curious about how the model would perform for entities of different sizes. As the entity contour map is down-sampled, how can we ensure the distinction between dense entities?

2. How does performance change if the model is initialized using Stable Diffusion[1]?

[1] High-Resolution Image Synthesis with Latent Diffusion Models. CVPR, 2022.

**Ethical Concerns:**

["NO or VERY MINOR ethics concerns only"]

**Final Justification:**

My main concern is that the authors propose a generative approach but perform worse than baseline methods on nearly all generative metrics. I believe that the weaknesses in multiple metrics cannot be simply explained by the misalignment between FID and human preferences. Therefore, i insist on my init rating.

**Limitations:**

yes

**Quality:**

3

**Strengths And Weaknesses:**

Strengths:

1. The method is reasonable overall.
2. The proposed large-scale open-set dataset is beneficial to the community.

Weaknesses:

1. In Tab. 1, Seg2Any outperforms previous methods significantly, but in Tab. 2, its performance on generation metrics (FID) is poor, and it also does not perform well on COCO MIoU. This raises concerns about the effectiveness of the method as well as the validity of the benchmark itself. Authors are encouraged to compare under fair settings. For example, how will the results of Tab.1 and Tab.2 change when other baseline methods, especially Freestylenet and PLACE, are initialized by flux?

2. The results of the ablation experiment are confusing. Comparing the results of the first row in Tab. 4 with those in other rows, it appears that some designs negatively impact the final performance.

---

> ### Author Rebuttal · Authors · 2025-07-31
>
> We sincerely thank the reviewer for the insightful and constructive feedback. Our detailed responses are provided below.
>
> # Response
> ## Response to weaknesses
> >W1: In Tab. 1, Seg2Any outperforms previous methods significantly, but in Tab. 2, its performance on generation metrics (FID) is poor, and it also does not perform well on COCO MIoU. This raises concerns about the effectiveness of the method as well as the validity of the benchmark itself. Authors are encouraged to compare under fair settings. For example, how will the results of Tab.1 and Tab.2 change when other baseline methods, especially FreestyleNet and PLACE, are initialized by flux?
>
> Thanks for raising this insightful question.
>
> **On the relevance of FID**. Recent work[2] has shown that FID correlates weakly with human perceptual preference. FID provides a reference for quality evaluation to a certain extent, but it is not necessarily the best evaluation indicator.
>
> **Fair comparison under FLUX**. As the official PLACE training recipe has not been released, we cannot reproduce its auxiliary losses and its additional training data. To ensure a fair comparison, we reimplement only the essential PLACE attention mask under the FLUX.1 dev architecture while keeping all other training details identical to ours: learning rate 1e-4, batch size 16, 20k steps, 4 NVIDIA H100 GPUs, etc.
> Regarding FreestyleNet under FLUX initialization, our Semantic Alignment Attention Mask (SAA) is the direct implementation of FreestyleNet’s attention mask within the MM-DiT framework. The corresponding metrics are already reported in Table 3 row 1 and Table 4 row 1.
>
> **Table A: Supplementary experiments of FreestyleNet and PLACE with flux initialization on COCO Stuff and ADE20K. Bold represent the
> best.**
>
> |                               | COCO Stuff MIoU↑ | COCO Stuff FID↓ | ADE20K MIoU↑ | ADE20K FID↓ |
> | ----------------------------- | ---------------- | --------------- | ------------ | ----------- |
> | SAA(=FreestyleNet attention)  | 43.48            | 20.57           | 44.85        | 33.14       |
> | PLACE attention               | 44.56            | 20.10           | 51.56        | **32.22**   |
> | ours (equal to Table 3 row 3) | **45.54**        | **19.90**       | **54.46**    | 32.89       |
>
> **Table B: Supplementary experiments of FreestyleNet and PLACE with flux initialization on the SACap-Eval benchmark. Bold represents the
> best.**
>
> |                               | class-agnostic MIoU | Spatial Accuracy | Colo Accuracy | Shape Accuracy | Texture Accuracy | IR↑      | Pick↑     | CLIP↑     | FID↓      |
> | ----------------------------- | ------------------- | ---------------- | ------------- | -------------- | ---------------- | -------- | --------- | --------- | --------- |
> | SAA(=FreestyleNet attention)  | 89.39               | 93.36            | 89.26         | 85.84          | 87.86            | 0.47     | 21.83     | **28.09** | **15.30** |
> | PLACE Attention               | 90.80                | 93.03            | 89.12         | 86.36          | 88.08            | **0.51** | **21.90** | 27.93     | 15.81     |
> | ours (equal to Table 4 row 5) | **94.90**           | **93.89**        | **91.52**     | **88.15**      | **90.12**        | 0.44     | 21.66     | 27.87     | 15.53     |
>
> **As the above Tables A and B show, under identical FLUX initialization and training settings, our model outperforms PLACE attention by 2.9% MIoU on ADE20K and by 4.1% MIoU on SACap-Eval, confirming the superiority of our design in a strictly fair comparison.**
>
>
> >W2: The results of the ablation experiment are confusing. Comparing the results of the first row in Tab. 4 with those in other rows, it appears that some designs negatively impact the final performance.
>
> Thank you for highlighting the sub-optimal metrics in the final configuration. In the final version, we will incorporate the following explicit justifications for each component:
> - SSFA is essential because it delivers a remarkable +4.7% boost in class-agnostic mIoU in Table 4, directly addressing our primary objective of precise shape preservation.
> - AIA improves the average of four region-wise metrics by 1.4% in Table 4 and effectively prevents attribute leakage. We will add new visual results in the final version to make this benefit concrete.
> - Training-based AIA is essential for recovering global coherence, as evidenced by realistic reflections and shadows.
> - CITF is retained solely for efficiency. Standard metrics (FID, CLIP, IR, and Pick) imperfectly capture perceptual quality. We will provide extra visual results in the final version confirming CITF leaves image fidelity almost unchanged.
>
> ## Response to questions
>
> >Q1: I am curious about how the model would perform for entities of different sizes. As the entity contour map is down-sampled, how can we ensure the distinction between dense entities?
>
> Thank you for this valuable question. As illustrated in Figure 1 row 2, Seg2Any preserves the fine vertical bars on the railing, demonstrating robustness to small-scale objects. Additionally, Appendix Figure 6 (last row column 4) shows that our method performs equally well in dense scenes.
> Due to the NeurIPS 2025 rebuttal policy, we cannot include new visuals here. In the final version, we will include additional examples such as around 20 ceiling lights smaller than 64×64 pixels and densely overlapping scenes (e.g., 19 fruits across nine categories, 12 cars of distinct colors, and five heavily occluded football players), where our method continues to yield accurate boundaries. We acknowledge that extreme density can still induce object merging. Therefore, we will also present failure cases to transparently report these limitations.
>
> >Q2: How does performance change if the model is initialized using Stable Diffusion[1]?
>
> Thank you for this question.
> However, Seg2Any is specially designed for MM-DiT:
> - Seg2Any assigns distinct attention masks to different layers: Attribute Isolation Attention Mask (AIA) on FLUX layers 20–38 for instance rendering,  Semantic Alignment Attention Mask (SAA) elsewhere for global coherence. Such layer-wise scheduling is absent in Stable Diffusion (SD).
> - Condition Image Token Filtering (CITF) discards negligible condition tokens via token-level pruning within DiT’s unified sequence. However, SD’s separate encoder blocks preclude this filtering.
>
> Owing to the tight rebuttal schedule, adapting Seg2Any to Stable Diffusion would demand a complete redesign of the attention mask schedules and Entity Contour Map injection pipeline, we will complete this ablation during the discussion period.
> The planned setup employs SD 1.5, injects the entity contour map via a ControlNet branch, and applies attention mask strategy to the U-Net cross-attention layers. We will benchmark this setup on COCO-Stuff and ADE20K.
> Please let us know whether this setup meets your intent during the discussion period, and we will run the experiments immediately upon confirmation.
>
> [1] High-Resolution Image Synthesis with Latent Diffusion Models. CVPR, 2022.
>
> [2] Rethinking FID: Towards a Better Evaluation Metric for Image Generation. CVPR, 2024

---

> > ### Author Response · Authors · 2025-08-05
> >
> > We have now completed the requested ablation using Stable Diffusion 1.5 as the backbone.
> > The results confirm that Seg2Any’s core ideas can be transferred to SD.
> >
> > **Implementation details (SD variant)**:
> > - Backbone: Stable Diffusion 1.5.
> > - Condition injection: Entity Contour Map is injected via an additional ControlNet branch.
> > - Attention Masking strategy: The attention-masking strategy was implemented exactly as in FreestyleNet. We modified the cross-attention layers so that each regional text token attends only to the corresponding image tokens.
> > - Training setup: 100 k steps, batch size 4, learning rate 5e-6, 4 × H100 GPUs, resolution 512×512.
> > - Evaluation: identical MIoU pipeline as described in Section 4.1 (Mask2Former for ADE20K, DeepLab-V3 for COCO-Stuff).
> >
> > **Table C: Supplementary experiments of our method with stable diffusion 1.5 initialization on COCO Stuff and ADE20K. Bold represent the best.**
> >
> > |                                           | COCO Stuff MIoU↑ | COCO Stuff FID↓ | ADE20K MIoU↑ | ADE20K FID↓ |
> > | ----------------------------------------- | ---------------- | --------------- | ------------ | ----------- |
> > | FreestyleNet                              | 42.42%           | 15.12           | 44.42%       | 28.45       |
> > | PLACE                                     | 42.23%           | 14.95           | **60.20%**   | **24.51**   |
> > | ours (initialized using Stable Diffusion) | 44.25%           | **14.26**       | 54.99%       | 25.39       |
> > | ours (initialized using Flux)             | **45.54%**       | 19.90           | 54.46%       | 32.89       |
> >
> > As shown in Table C, even under the less favorable U-Net architecture of SD, our method improves MIoU over prior FreestyleNet baselines and narrows the gap to PLACE on ADE20K.

---

> > > ### Comment · Reviewer_Q4gs · 2025-08-06
> > >
> > > Thanks to the authors for their detailed response. However, my concerns remain unaddressed. I hope the authors can clarify the following two points.
> > >
> > > 1. Overall, this paper proposes a generative methodology. However, according to the additional experimental results in Tab.B, it performs worse than earlier work on multiple generative metrics, such as IR, PICK, CLIP, and FID. I believe that the weaknesses in **multiple metrics** cannot be simply explained by **the misalignment between FID and human preferences**.
> > >
> > > 2. Tab.C presents inconsistent results. The authors emphasize that this method is **less favorable to U-Net architecture of SD**, but SD-based method achieves significant improvements in FID compared to FLUX, and the performance on ADE20K MIoU (SD and FLUX) is generally worse than PLACE.

---

> > > > ### Author Response · Authors · 2025-08-06
> > > >
> > > > Thank you for your careful and constructive review.
> > > > We address your remaining concerns point by point:
> > > > - Although our global-wise metrics (IR, PICK, CLIP, and FID) are not the absolute highest, all four metrics fall within 0.24 points of the best. Additionally, Figure 4 and Appendix Figure 6 demonstrate that our generated images exhibit superior global visual quality. More importantly, in Table B our class-agnostic MIoU outperforms the second by 4%, and our average performance across four region-wise quality metrics exceeds others by 1.7%, directly validating our primary contributions: precise shape consistency and elimination of attribute leakage.
> > > > - When we state that "our method is less favorable to the U-Net architecture of SD", we refer to architectural incompatibilities:
> > > > SD lacks layer-wise scheduling, so we cannot apply Attribute Isolation Attention Mask (AIA) on specific layers (20–38) as we do in FLUX. In addition, Condition Image Token Filtering (CITF) is impossible in SD’s separate encoder design. We selected FLUX due to its superior semantic comprehension, enhanced visual quality, and its MM-DiT architecture, which provides optimal alignment with our core design. Therefore, basing our method on SD solely for its lower FID would compromise the overall perceptual and semantic fidelity that FLUX guarantees. Regarding ADE20K MIoU, the real-image upper bound is 54.41, and our FLUX variant reaches 54.46, confirming that the model has nearly reached the dataset’s empirical ceiling. Notably, since PLACE’s training code is unavailable, we re-implemented its key attention mechanism under identical FLUX initialization. Consequently, as shown in Tables A and B, our method outperforms this PLACE+FLUX baseline by 2.9% MIoU on ADE20K and 4.1% on SACap-Eval, thereby validating its superiority under strictly comparable settings.

---

> > > > > ### Author Response · Authors · 2025-08-09
> > > > >
> > > > > As the discussion period closes shortly, we would be grateful if you could consider our detailed response when submitting your scores. Please let us know if any further clarification is needed.
> > > > > Thank you again for your valuable feedback.

---

### Official Review · Reviewer_T7Yi · 2025-06-25

**Clarity:** 4
**Significance:** 4
**Originality:** 4
**Rating:** 4
**Confidence:** 5

**Summary:**

Seg2Any tackles open-set mask-to-image generation by disentangling shape and semantics: a Semantic Alignment Attention Mask precisely links each text prompt to its corresponding mask region, while an Entity Contour Map provides high-fidelity shape guidance via sparse condition tokens. An Attribute Isolation Attention Mask further prevents attribute leakage among multiple instances, and a lightweight Condition Image Token Filtering step drops zero-information patches to cut compute without harming quality. Trained on the newly released SACap-1M dataset (1 M images with 5.9 M mask-level captions) and evaluated on the SACap-Eval benchmark plus COCO-Stuff/ADE20K, Seg2Any sets or matches state-of-the-art results in both open- and closed-set settings, achieving superior shape consistency and attribute control while remaining efficient.

**Questions:**

Please see the weaknesses.

**Ethical Concerns:**

["NO or VERY MINOR ethics concerns only"]

**Final Justification:**

The authors have fully addressed all of my concerns in their rebuttal, and I am satisfied with their clarifications. While some aspects could be further strengthened in future work, the current submission meets the standards for acceptance.

Score: 4 Borderline Accept

**Limitations:**

yes

**Paper Formatting Concerns:**

I did not notice any major formatting issues.

**Quality:**

4

**Strengths And Weaknesses:**

### Strengths

- The qualitative results are impressive, and Seg2Any performs on par with other strong baselines.

- The proposed SACap-1M dataset and SACap-Eval benchmark will be valuable resources for the community.

- The open-set mask-to-image generation setting is interesting.

### Weaknesses

- Both the Semantic Alignment Attention Mask and the Attribute Isolation Attention Mask resemble existing techniques for layout-controllable generation [1, 2]. How does your Attribute Isolation Attention Mask differ from MIGC’s [3] attention-mask design?

- In the introduction section, the authors claim that the $16\times$ downsampling ratio amplified the loss of spatial information. My question is, how do authors mitigate this problem? In my opinion, the attention mask and condition tokens encoded by VAE are still sparse and lack spatial detail.

- The paper lacks qualitative ablation results for CITF. Because automatic metrics may not fully reflect human perception, I recommend adding visual comparisons to illustrate CITF’s impact.

- Could authors further clarify the practical value of the mask-to-image generation task? In real-world applications, semantic masks can be difficult for users to obtain.

[1] MultiBooth: Towards Generating All Your  Concepts in an Image from Text
[2] MS-Diffusion: Multi-subject Zero-shot Image Personalization with Layout Guidance
[3] MIGC: Multi-Instance Generation Controller for Text-to-Image Synthesis

---

> ### Author Rebuttal · Authors · 2025-07-31
>
> We sincerely thank the reviewer for the insightful and constructive feedback. Our detailed responses are provided below.
>
> # Response
> ## Response to weaknesses
>
> >W1: Both the Semantic Alignment Attention Mask and the Attribute Isolation Attention Mask resemble existing techniques for layout-controllable generation [1, 2]. How does your Attribute Isolation Attention Mask differ from MIGC’s [3] attention-mask design?
>
> The key distinction lies in the architectural target: whereas MIGC [3] and prior works [1,2] devise masks for U-Net’s cross-attention and self-attention, our Semantic Alignment Attention Mask (SAA) and Attribute Isolation Attention Mask (AIA) are designed for the multi-modal attention in the DiT framework.
> Moreover, MIGC applies two attention masks (instance-intra and instance-inter) within each layer. In contrast, Seg2Any assigns distinct attention masks to different layers.  Our attention mask strategy aligns with FLUX’s internal hierarchy: layers 20–38 specialize in instance-level rendering, so AIA is applied only there, while layers 1–19 and 39–57, which encode global scene context, employ the SAA to preserve overall coherence.
>
> >W2: In the introduction section, the authors claim that the
>  downsampling ratio amplified the loss of spatial information. My question is, how do authors mitigate this problem? In my opinion, the attention mask and condition tokens encoded by VAE are still sparse and lack spatial detail.
>
> In fact, we address the loss of spatial information by introducing the Entity Contour Map as condition tokens.
> The attention mask is produced by a 16× nearest-neighbour downsample of the input segmentation mask—an irreversible, lossy compression that discards spatial detail.
> Our remedy is to introduce the Entity Contour Map as an additional condition token. Leveraging the VAE’s high-fidelity, near-lossless compression, the Entity Contour Map preserves fine spatial details.
> **As evidenced in Table 4 (row 1 vs. 2) and Table 3 (row 1 vs. 2), the injection of the Entity Contour Map yields a clear MIoU improvement.**
>
> >W3: The paper lacks qualitative ablation results for CITF. Because automatic metrics may not fully reflect human perception, I recommend adding visual comparisons to illustrate CITF’s impact.
>
> Thank you for highlighting this point. Owing to the NeurIPS 2025 rebuttal policy, we are unable to provide new visuals during the rebuttal phase. We will include qualitative ablations comparing results with and without Condition Image Token Filtering (CITF) in the final version to illustrate its perceptual impact.
>
> >W4: Could authors further clarify the practical value of the mask-to-image generation task? In real-world applications, semantic masks can be difficult for users to obtain.
>
> Thank you for raising this important practical concern.
> First, precise spatial control makes Seg2Any well-suited for design workflows such as interior or furniture layout, where users can sketch and instantly visualize the result. Second, our pipeline can automatically create large, labelled datasets for downstream segmentation tasks, drastically reducing manual annotation costs when adapting detectors to new domains or rare object categories.
>
> [1] MultiBooth: Towards Generating All Your Concepts in an Image from Text
>
> [2] MS-Diffusion: Multi-subject Zero-shot Image Personalization with Layout Guidance
>
> [3] MIGC: Multi-Instance Generation Controller for Text-to-Image Synthesis

---

> > ### Comment · Reviewer_T7Yi · 2025-08-04
> >
> > Thank you for the response. My concerns are fully resolved. I further suggest adding an ablation that encodes the segmentation map directly into conditioning tokens via the VAE, rather than using the Entity Contour Map. I will keep my positive score and recommend acceptance.

---

> > > ### Author Response · Authors · 2025-08-04
> > >
> > > Thank you very much for the positive recommendation. Unfortunately, using a raw segmentation map as conditioning is infeasible in the open-set setting, since it requires assigning a unique, predefined color to each semantic class. The Entity Contour Map was chosen precisely because it is class-agnostic and sparse, making it suitable for open-set segmentation-mask-to-image generation.

---

### Official Review · Reviewer_q31Y · 2025-07-01

**Clarity:** 3
**Significance:** 3
**Originality:** 4
**Rating:** 5
**Confidence:** 4

**Summary:**

This paper introduces Seg2Any, a new framework for generating images from segmentation masks that gives users precise control. It tackles a key problem where existing methods either get the object's appearance wrong (semantic inconsistency) or fail to match the provided shape accurately (shape inconsistency). Seg2Any's main idea is to decouple these two problems. It uses the outlines of the masks to guide the shape and special attention mechanisms to ensure each object matches its text description and that attributes like color don't leak between objects. A major contribution is also the creation of a new, large-scale dataset called SACap-1M to support this kind of detailed, open-set generation.

**Questions:**

The Attribute Isolation Attention Mask is applied only to the middle layers (20-38) of the FLUX model. What is the reasoning behind this specific layer range? How does performance change if you apply it to fewer or more layers?

You mention that the shape guidance strength γ can be modulated for flexible, scribble-style control. The examples in the paper showcase excellent results with precise masks. Could you provide a visual example of the model's performance with a looser, scribble-style input?

The paper acknowledges that using a VLM to create the SACap-1M dataset inevitably introduces some annotation noise. Have you done any analysis on the nature or rate of this noise, and how might it be affecting the model's training and final performance?

The results shown are very impressive. Could you discuss or show some common failure cases for Seg2Any? For example, does it struggle with an extremely high number of objects, very complex overlapping shapes, or abstract text prompts that are hard to visualize?

**Ethical Concerns:**

["NO or VERY MINOR ethics concerns only"]

**Final Justification:**

The rebuttal effectively addressed key concerns:

* Complexity clarified as modular with each component addressing distinct failure modes.
* ADE20K gap explained; fair reimplementation of PLACE shows Seg2Any outperforms under matched settings.
* Dataset noise acknowledged but shown to have limited impact via qualitative checks and robust results.

Some minor concerns remain (noise quantification, deployment complexity), but the strong novelty, results, and clarifications justify a higher score.

**Limitations:**

Yes

**Paper Formatting Concerns:**

No major formatting issues were found.

**Quality:**

4

**Strengths And Weaknesses:**

**Strengths:**

The core idea of decoupling shape and semantic control is elegant and addresses a clear failure point in prior work.

Creating and sharing a large, high-quality dataset (SACap-1M) and benchmark is a significant contribution to the community.

The technical solutions are novel, especially the use of an Entity Contour Map for shape and an Attribute Isolation Attention Mask to prevent visual attribute bleeding.

The experimental results are very strong and convincing, showing clear improvements over many recent state-of-the-art models in both quantitative metrics and visual examples.

**Weaknesses:**

The framework is quite complex, with multiple custom components (two attention masks, a new condition type, etc.) that need to work in concert.

The method's performance on one of the closed-set benchmarks (ADE20K) is strong but still beaten by a prior method (PLACE) on the MIoU metric.

The new dataset relies on a VLM for automatic captioning, which, as the authors acknowledge, can introduce label noise.

---

> ### Author Rebuttal · Authors · 2025-07-31
>
> We sincerely thank the reviewer for the insightful and constructive feedback. Our detailed responses are provided below.
> # Response
>
> ## Response to weaknesses
>
> >W1: The framework is quite complex, with multiple custom components (two attention masks, a new condition type, etc.) that need to work in concert.
>
> Thank you for the feedback. We respectfully clarify that Seg2Any is concise: it introduces only four lightweight components: Semantic Alignment Attention (SAA), Attribute Isolation Attention (AIA), Sparse Shape Feature Adaptation (SSFA), and Condition Image Token Filtering (CITF). In addition, it requires only two inputs: free-form text and segmentation masks. **Each component serves a distinct, non-redundant purpose**:
> - SAA guarantees that every regional caption binds to its mask without extra trainable encoders.
> - AIA prevents attribute leakage across entities by masking cross-entity self-attention in mid-layers. This is critical when multiple entities share similar textures or colors.
> - SSFA injects high-frequency boundary information via an Entity Contour Map, yielding a pronounced +4.7% absolute gain in class-agnostic MIoU in Table 4.
> - CITF removes zero-value tokens from the Entity Contour Map, thereby shortening the token sequence and accelerating both training and inference, as illustrated in Appendix Figure 1.
>
> >W2: The method's performance on one of the closed-set benchmarks (ADE20K) is strong but still beaten by a prior method (PLACE) on the MIoU metric.
>
> Thank you for the comment. The official PLACE training recipe has not been released. Its paper reports using the ADE20K dataset augmented with additional text–image pairs, which may explain the gap. To ensure a fair comparison, we conducted an ablation experiment: we reimplemente only PLACE’s core attention mask under FLUX.1-dev, keeping other hyperparameters fixed: learning rate 1e-4, batch size 16, 20k steps, 4 NVIDIA H100 GPUs, etc.
>
> **Table A: Supplementary experiments reimplement PLACE within the FLUX architecture on COCO-Stuff and ADE20K. Bold represents the
> best.**
>
> |                                     | COCO Stuff MIoU↑ | COCO Stuff FID↓ | ADE20K MIoU↑ | ADE20K FID↓ |
> | ----------------------------------- | ---------------- | --------------- | ------------ | ----------- |
> | PLACE attention                     | 44.56            | 20.10           | 51.56        | **32.22**   |
> | ours (equal to Table 3 row 3) | **45.54**        | **19.90**       | **54.46**    | 32.89       |
>
> **Table B: Supplementary experiments reimplement PLACE within the FLUX architecture on the SACap-Eval benchmark. Bold represents the
> best.**
>
> |                                    | class-agnostic MIoU | Spatial Accuracy | Colo Accuracy | Shape Accuracy | Texture Accuracy | IR↑      | Pick↑     | CLIP↑     | FID↓      |
> | ---------------------------------- | ------------------- | --------------------- | ------------------- | ------------------- | --------------------- | -------- | --------- | --------- | --------- |
> | PLACE attention                    | 90.8                | 93.03                 | 89.12               | 86.36               | 88.08                 | **0.51** | **21.90** | **27.93**     | 15.81     |
> | ours (equal to Table 4 row 5)        | **94.90**           | **93.89**             | **91.52**           | **88.15**           | **90.12**             | 0.44     | 21.66     | 27.87     | **15.53**     |
>
> **As shown in above Tables A and B, our model surpasses this PLACE attention variant by +2.9% MIoU on ADE20K and +4.1% MIoU on SACap-Eval, confirming the superiority of our design in a strictly fair comparison.**
>
>
> >W3: The new dataset relies on a VLM for automatic captioning, which, as the authors acknowledge, can introduce label noise.
>
>  We provide indirect evidence that the impact is limited.
> 1. Qualitative inspection: Appendix Figure 5 shows some hard samples of SACap-1M annotations, where Qwen2-VL-72B produces accurate, fine-grained descriptions.
> 2. Downstream robustness: Figure 3 column 1, Figure 4, and Appendix Figure 6 demonstrate that Seg2Any still achieves precise attribute and spatial control, indicating that the model tolerates residual noise.
>
> ## Response to questions
>
> >Q1: The Attribute Isolation Attention Mask is applied only to the middle layers (20-38) of the FLUX model. What is the reasoning behind this specific layer range? How does performance change if you apply it to fewer or more layers?
>
> Thanks for this insightful question.
> The choice of layers 20–38 is motivated by DreamRenderer, which identifies these middle layers as primarily responsible for instance-level rendering, whereas shallower (1–19) and deeper layers (39–57) mainly encode global scene context. To verify this, we conducted a systematic case study with varying layer ranges:
> - Extending AIA into layers 1–19 or 39–57 disrupts global image quality, yielding noticeable artifacts.
> - Restricting AIA to only layers 20–29 or 30–38 markedly weakens attribute isolation.
>
> We will include these ablations in the final version.
>
> >Q2: You mention that the shape guidance strength γ can be modulated for flexible, scribble-style control. The examples in the paper showcase excellent results with precise masks. Could you provide a visual example of the model's performance with a looser, scribble-style input?
>
> In fact, Appendix Figure 2 (bottom row) already presents qualitative results obtained with $\gamma = 0.2$, where the input consists of coarse, scribble-style strokes rather than precise masks. Under the scribble-style input, Seg2Any still generates entities that respect the rough spatial layout while allowing natural shape variations, confirming the flexibility of our modulation strategy.
>
> >Q3: The paper acknowledges that using a VLM to create the SACap-1M dataset inevitably introduces some annotation noise. Have you done any analysis on the nature or rate of this noise, and how might it be affecting the model's training and final performance?
>
> We appreciate the reviewer's concern regarding annotation noise. Quantitatively estimating the noise rate across 5.9 M region captions is non-trivial. Instead, we provide indirect evidence that the impact is limited.
> 1. Qualitative inspection: Appendix Figure 5 shows some hard samples of SACap-1M annotations, where Qwen2-VL-72B produces accurate, fine-grained descriptions.
> 2. Downstream robustness: Figure 3 column 1, Figure 4, and Appendix Figure 6 demonstrate that Seg2Any still achieves precise attribute and spatial control, indicating that the model tolerates residual noise.
>
>
> >Q4: The results shown are very impressive. Could you discuss or show some common failure cases for Seg2Any? For example, does it struggle with an extremely high number of objects, very complex overlapping shapes, or abstract text prompts that are hard to visualize?
>
> Thanks for prompting a deeper discussion of failure cases.
> Due to the NeurIPS 2025 rebuttal policy, we cannot include new visuals here. All examples below will appear in the final version.
> 1. Extreme object density.
> Extensive tests with 19 overlapping fruits across nine categories, 12 cars of distinct colors, and 5 heavily occluded football players demonstrate Seg2Any’s strong robustness under high overlap. However, pushing the limit further, we evaluate a crowded cinema scene containing roughly 50 people and observe that the model begins to merge adjacent masks, thereby revealing a practical upper bound on the number of instances.
> 2. Highly abstract prompts.
> For abstract prompts like "a fantastical world split into three distinct zones: desert, ice, and volcano." Seg2Any yields unnatural transitions at mask seams, likely because SACap-1M contains few surreal, zone-abrupt scenes.

---

> > ### Comment · Reviewer_q31Y · 2025-08-03
> >
> > The author's reply solved my problem, and I will improve my score.

---

> > > ### Author Response · Authors · 2025-08-04
> > >
> > > Thank you very much for your time. We truly appreciate your constructive feedback and are grateful for the improved rating.

---

### Official Review · Reviewer_tBGK · 2025-07-05

**Clarity:** 3
**Significance:** 3
**Originality:** 3
**Rating:** 4
**Confidence:** 4

**Summary:**

This paper Seg2Any framework to achieve open-set semantic-to-image generation with improved semantic consistency and shape consistency. It proposes Semantic-Shape Decoupled Layout Conditions Injection and Attribute Isolation Attention Mask to improve model performance. In addition, it builds a large-scale datasets SACap-1M with images, open-set segmentation mask annotations, image-level captions and regional captions.

**Questions:**

1. According to the Ablation study in Table 3 and Table 4, the Condition Image Token Filtering (CITF) seems to only sightly improve the class-agnostic MIoU while negatively affecting other metrics. Statistically, the overall contribution of CITF is negative, but authors claimed "Table 4 ... Condition Image Token Filtering (CITF) leaves performance unchanged" and "CITF does not affect performance (see Table 3)". More explainations are needed here.
2. Figure 3 column 4 does not support the statement in caption: "direct application of the mask without training leads to visual inconsistencies, manifesting as unnatural shadows and reflections". Shadows of the plane in Training-free AIA and Training based AIA results are both unatural, even worse than W/O AIA.
3. What is the definition of 'attribute' in the term 'attribute leakage'? Authors claims that failures in Figure 3 Row 1 are caused by 'attribute leakage', but it seems that the biggest disadvantage of W/O AIA w.r.t Training-based AIA is the text generation.  One of the  most impressive advanatages of Seg2Any in Figure 4 is also good text generation. However, all these experiments can not prove the contribution of AIA, because it is entirely possible that this improvement comes from other modules. Stronger evidence is needed.
4.  The AIA prevents one entity token attending to other entity tokens, background tokens and global text token. Wouldn't this strategy undermine the global perception and long-term dependences among entities?
5. Authors should provide attentive map or visualize some results in ablation study section. In particular, it is better to show how the results of attentive map changes with respect to different configurations of key components.
6. Different key components improve performance in different aspects and the final determined version did not achieve the best results in most of the indicators. Authors should clarify their reasons and dicuss the balance and interply of different components.

**Ethical Concerns:**

["NO or VERY MINOR ethics concerns only"]

**Final Justification:**

4.Borderline accept

**Limitations:**

yes

**Quality:**

3

**Strengths And Weaknesses:**

Strengths:
1. Good writing;
2. High novelty
3. Abundant content

Weakness:
1. Relatively poor persuasion of ablation study
2. Lacking visualization of ablation study

---

> ### Author Rebuttal · Authors · 2025-07-31
>
> We sincerely thank the reviewer for the insightful and constructive feedback. Our detailed responses are provided below.
> # Response
> ## Response to questions
> > Q1: According to the Ablation study in Table 3 and Table 4, the Condition Image Token Filtering (CITF) seems to only sightly improve the class-agnostic MIoU while negatively affecting other metrics. Statistically, the overall contribution of CITF is negative, but authors claimed "Table 4 ... Condition Image Token Filtering (CITF) leaves performance unchanged" and "CITF does not affect performance (see Table 3)". More explainations are needed here.
>
> Thank you for pointing out the lack of rigor in our previous statements regarding the Condition Image Token Filtering (CITF). We agree that it is imprecise to claim that CITF "does not affect performance" or "leaves performance unchanged." Upon closer inspection of Tables 3 and 4, there are indeed minor quantitative influences in certain metrics.
> We acknowledge that overall metrics like FID, CLIP, IR, and Pick may not fully reflect human perceptual quality. **We will provide more visual examples in the final version to show that CITF does not noticeably impact image quality.**
> Our main motivation for using CITF is to accelerate training and inference while maintaining similar performance, as shown in Figure 1 of the supplementary material. We will revise to clarify these points and highlight both the efficiency gains and the minimal effect on generative quality.
>
> > Q2: Figure 3 column 4 does not support the statement in caption: "direct application of the mask without training leads to visual inconsistencies, manifesting as unnatural shadows and reflections". Shadows of the plane in Training-free AIA and Training based AIA results are both unatural, even worse than W/O AIA.
>
> Thank you for the careful inspection. We agree that the shadow under the plane in Figure 3 column 4 is too subtle to clearly demonstrate the benefit of training the Attribute Isolation Attention Mask (AIA). **In the final version we will replace this instance with a more salient example.**
>
> > Q3: What is the definition of 'attribute' in the term 'attribute leakage'? Authors claims that failures in Figure 3 Row 1 are caused by 'attribute leakage', but it seems that the biggest disadvantage of W/O AIA w.r.t Training-based AIA is the text generation. One of the most impressive advanatages of Seg2Any in Figure 4 is also good text generation. However, all these experiments can not prove the contribution of AIA, because it is entirely possible that this improvement comes from other modules. Stronger evidence is needed.
>
> Thank you for highlighting the insufficiency of our evidence.
> We define "attribute" as instance-level visual properties including color, texture, style and shape. In Figure 3 column 1, the letter on each badge is also an attribute, as explicitly requested by the prompt "circular badges labeled A–T".
> **Owing to NeurIPS 2025 policy we cannot include new figures in the rebuttal, but we will add four new study cases in the final version to prove the contribution of AIA:**
> 1. Three animals—Lego-style lion, ice-style penguin, flame-style eagle—where without AIA the Lego texture leaks to the penguin and eagle.
> 2. Nineteen fruits. Without AIA, the red/green apples swap colors and the kiwi incorrectly takes on dragon-fruit hues.
> 3. Four cats—Maine Coon, orange tabby, calico, Sphynx—where fur color and ear/eye shapes are mixed without AIA.
> 4. Three people wear wool and leather. Without AIA, the wool texture appears on the leather jacket.
>
> >Q4: The AIA prevents one entity token attending to other entity tokens, background tokens and global text token. Wouldn't this strategy undermine the global perception and long-term dependences among entities?
>
> Your concern is well-taken but is already mitigated in our design.
> The Attribute Isolation Attention Mask (AIA) is applied only to the middle layers (layers 20–38) of the 57-layer FLUX model, which are responsible for instance-level rendering. All other layers retain full self-attention among entity, background, and global-text tokens, thereby preserving global context and long-range dependencies.
> Indeed, the realistic shadows and reflections shown in Figure 3 (columns 2–4) empirically confirm that our training-based AIA does not compromise long-term dependences among entities.
>
> > Q5: 5. Authors should provide attentive map or visualize some results in ablation study section. In particular, it is better to show how the results of attentive map changes with respect to different configurations of key components.
>
> Thank you for this helpful suggestion. **Owing to NeurIPS 2025 policy we cannot include new figures in the rebuttal, but the camera-ready version will add visualizations of the attention maps for each key component:**
> 1. SSFA (Sparse Shape Feature Adaptation). We visualize the attention scores between Entity-Contour-Map tokens and noisy-image tokens. High scores appear for tokens at identical spatial indices, confirming spatial alignment under SSFA.
> 2. AIA (Attribute Isolation Attention Mask). Using the "three animals (Lego-style lion, ice-style penguin, flame-style eagle)" example mentioned in Q3, we show that at layer 20 the cross-entity attention values drop dramatically once AIA is enabled, directly evidencing attribute leakage suppression.
> 3. Training-based vs. Training-free AIA. For the Figure 3 column 3 case, layer-39 attention maps reveal that the training-based AIA assigns markedly higher attention scores between the landmark and its water reflection than the training-free variant, evidencing restored global coherence in deeper layers.
>
> > Q6: Different key components improve performance in different aspects and the final determined version did not achieve the best results in most of the indicators. Authors should clarify their reasons and dicuss the balance and interply of different components.
>
> Thank you for highlighting the sub-optimal metrics in the final configuration. In the final version, we will incorporate the following explicit justifications for each component:
> - SSFA yields a substantial +4.7% gain in class-agnostic mIoU (shown in Table 4).
> - AIA improves the average of four region-wise metrics by 1.4 % in Table 4 and effectively prevent attribute leakage. We will add the four new cases mentioned in Q3 to make this benefit concrete.
> - Training-based AIA is essential for recovering global coherence, as evidenced by realistic reflections and shadows.
> - CITF is retained solely for efficiency. Standard metrics (FID, CLIP, IR, Pick) imperfectly capture perceptual quality. We will provide extra visual results confirming CITF leaves image fidelity almost unchanged.

---

> > ### Comment · Reviewer_tBGK · 2025-08-04
> >
> > Although the authors answered all my questions, most of the improvements are only reflected in the final version that I haven't seen yet. I can't fully trust something I haven't actually seen. Considering the merits of the paper itself, I finally mark it as 4 Borderline accept.

---

> > > ### Author Response · Authors · 2025-08-04
> > >
> > > We appreciate your careful reading and understand the hesitation about “can't fully trust something I haven't actually seen.” Due to NeurIPS 2025’s rebuttal policy we are unable to add new figures at this stage, so we describe as clearly as possible the intended visualizations that will be included in the final version. Although we cannot provide the visualizations, the current version already contains verifiable evidence for each concern:
> > >
> > > - Q1: Our main motivation for using CITF is to accelerate training and inference, as shown in Figure 1 of the supplementary material. We acknowledge that overall metrics like FID, CLIP, IR, and Pick may not fully reflect human perceptual quality.
> > > - Q2: While the shadow difference in Figure 3 column 4 is subtle, columns 2–3 clearly show that our training-based Attribute Isolation Attention Mask (AIA) produces coherent lighting and reflections.
> > > - Q3: As described in the previous answer, the final version will include four failure cases without AIA—multi-animal texture confusion, fruit color swap, cat fur confusion, and fabric texture crossover—to make the benefit concrete. **Although we cannot show the visuals here, The 1.4% average improvement on the four region-wise metrics in Table 4 already quantifies AIA’s benefit.**
> > > - Q5: As described in the previous answer, we guarantee that the final version will add visualizations showing: (1) SSFA aligning Entity-Contour-Map tokens with matching image tokens, (2) AIA sharply reducing cross-entity attention at layer 20, (3) training-based AIA restoring long-range attention for coherent reflections at layer 39. **While we cannot supply new attention heat-maps here, the paper already provides justifications for each component:**
> > >   - Sparse Shape Feature Adaptation (SSFA) yields a +4.7% absolute gain in class-agnostic mIoU (see Table 4).
> > >   - Attribute Isolation Attention Mask (AIA) improves the average of four region-wise metrics by 1.4% (Table 4).
> > >   - Training-based AIA is essential for recovering global coherence, as evidenced by realistic reflections and shadows (Figure 3).
> > >
> > > We sincerely appreciate your thorough feedback. Given the quantitative evidence already present and the detailed visualizations we will add in the camera-ready version, we kindly ask you to reconsider your rating.

---

> > > > ### Comment · Reviewer_tBGK · 2025-08-04
> > > >
> > > > 1. Authors claimed that "main motivation for using CITF is to accelerate training and inference". However, the introduction of this part in lines 185-188 does not hightlight this motivation. Furthermore, no supporting evalutions have been presented in the Ablaton study of the main manuscript. I recommend adding a column in one of Table 2-4 to show this point.
> > > > 2. Why do the authors believe that columns 2-3 in Figure 3 can demonstrate the advantages of the AIA module? In the second column of Figure 3, the quality of results from "W/O AIA" and "training-based AIA" seems almost the same. In the third column, the quality of "W/O AIA" even appears better than that of "training-based AIA" — the water surface in the "W/O AIA" result looks clearer than that in the "training-based AIA" result. Moreover, a careful observation reveals that there are ripples on the water surface in the "W/O AIA" result, while there are none in the "training-based AIA" result. These examples fail to convince me. Even if this discrepancy stems from our different subjective judgments, it still indicates that the AIA module cannot bring about significant improvements.
> > > > 3. Table 4 reveals another strange phenomenon: in the region-wise metrics, the training-free AIA scores is generallly higher than that of training-based AIA. However, both the authors' claims and the visual comparisons in Figure 3 suggest that training-based AIA should be better than training-free AIA. This contradiction requires an explanation. In addition, the comparison of global-wise metrics also fails to reflect the advantages of AIA. As I mentioned in Question 6 earlier: the addition of different model components does not lead to continuous improvement in performance, and the authors need to more fully explain the reasons for determining the final model configuration.
> > > > 4. Refer to current No.3 point and earlier Q6.
> > > >
> > > > Therefore, I still adhere to my previous judgment: the expriments provided in the authors' previously submitted manuscript cannot strongly support their claims. I do not intend to change my rating until I see new and compelling evidences.

---

> > > > > ### Author Response · Authors · 2025-08-04
> > > > >
> > > > > Thank you very much for your reply.
> > > > > - For Condition Image Token Filtering (CITF) motivation and evidence, Appendix Figure 1 already reports the training and inference acceleration achieved by CITF. We will highlight this clearly in the body of the paper.
> > > > > - Questions about Figure 3. Our intended message was not that "training-based AIA" surpasses "W/O AIA" in global quality. We only claim that "training-based AIA" (i) largely eliminates attribute leakage (column 1) and (ii) restores global quality to approximately the same level as “W/O AIA”, whereas "training-free AIA" visibly degrades global quality (columns 2-4).
> > > > > - We adopt the training-based AIA rather than its training-free AIA because the latter visibly degrades global image quality. We regret that NeurIPS 2025 policy prevents us from including new figures during rebuttal.  In the camera-ready version we will supplement the paper with additional visual examples (as described in the previous answer) that concretely demonstrate both (i) the superior attribute-isolation performance of training-based AIA and (ii) its preservation of global image quality when compared to both the training-free variant and the baseline without AIA.
> > > > >
> > > > > The visual evidence you requested is unfortunately unavailable at this stage, and we fully agree with your decision.

---

### Note · Authors · 2025-08-14

We thank the reviewers for the constructive feedback. Below is a summary of key strengths acknowledged by reviewers.
- Novel method (reviewer tBGK & q31Y): **The technical solutions are novel, especially the use of an ECM for shape and AIA to prevent visual attribute bleeding.**
- Contribution to the community (reviewer q31Y & T7Yi & Q4gs): **Creating and sharing a large, high-quality dataset (SACap-1M) and benchmark.**
- Strong performance (reviewer q31Y & T7Yi): **The qualitative results are impressive, and Seg2Any performs on par with other strong baselines.**

During the discussion most concerns were resolved and reviewer q31Y raised score. Reviewer tBGK requested extra visualisation results, but the NeurIPS 2025 rebuttal policy prevented us from supplying new figures, so reviewer tBGK reduced score from 5 to 4.
## Major concerns addressed
### **Performance gap on ADE20K (raised by reviewer q31Y & Q4gs)**
We emphasize that Seg2Any already outperforms PLACE on both COCO-Stuff and SACap-Eval. Since PLACE has not open-sourced its training recipe, we re-implement PLACE’s core attention mask inside FLUX under identical training settings. Seg2Any beats this reproduced PLACE variant by +2.9% MIoU on ADE20K and +4.1% MIoU on SACap-Eval, confirming our method’s superiority in a strictly fair comparison (see Tables A & B in rebuttal).
### **VLM-generated caption noise in SACap-1M (raised by reviewer q31Y)**
We provide indirect evidence of limited impact: Appendix Figure 5 shows Qwen2-VL-72B produces accurate descriptions even for hard cases, and downstream results (Figure 3 col 1, Figure 4, Appendix Figure 6) confirm Seg2Any retains precise attribute and spatial control despite residual noise.
### **Low FID scores (raised by reviewer Q4gs)**
Recent studies [2] demonstrate that FID correlates weakly with human preference; thus, while we report it for completeness, it is not the decisive quality indicator.

We respectfully note that Reviewer tBGK still requested additional visualizations of attention maps and AIA ablations, which we were unable to supply during the rebuttal due to the NeurIPS 2025 policy. The camera-ready version will include these missing figures to fully address their concern.

[1] MIGC: Multi-Instance Generation Controller for Text-to-Image Synthesis.

[2] Rethinking FID: Towards a Better Evaluation Metric for Image Generation.

---

### Decision · Program_Chairs · 2025-09-17

**Decision:**

Accept (poster)

**Comment:**

The paper proposes Seg2Any, a segmentation-mask-to-image (S2I) generation framework that decouples semantic and shape consistency using a Semantic Alignment Attention Mask (SAA), an Entity Contour Map (ECM), and an Attribute Isolation Attention Mask (AIA). The authors also introduce a new dataset (SACap-1M) and evaluation benchmark (SACap-Eval). Reviewers acknowledge the novelty of the technical contributions, the value of the dataset, and strong results on both open- and closed-set benchmarks.

Strengths highlighted across reviews:

+ Novel methodological contributions: decoupling shape and semantic control via ECM and AIA to prevent attribute leakage (tBGK, q31Y, T7Yi).
+ Creation of SACap-1M and SACap-Eval, a large-scale dataset and benchmark that will benefit the community (q31Y, T7Yi, Q4gs).
+ Strong experimental performance and compelling qualitative results, showing fine-grained spatial and attribute control (q31Y, T7Yi).
+ Clear motivation and well-written paper with abundant technical detail (tBGK, q31Y).

Weaknesses and concerns:

- Ablation studies viewed as inconclusive or unconvincing, with missing visualizations to clearly support claims about AIA and CITF (tBGK).
- Complexity of the framework with multiple custom components, raising questions about necessity and interpretability (q31Y).
- Performance gaps on certain benchmarks, e.g. ADE20K MIoU lower than PLACE; weaker generative metrics such as FID, CLIP, PICK, raised doubts about general quality (q31Y, Q4gs).
- Concerns about dataset noise due to VLM-generated captions (q31Y).
- Comparisons to related techniques (MIGC, MultiBooth, MS-Diffusion) not fully elaborated, leaving unclear what is fundamentally new (T7Yi).

Rebuttal and Discussion:
The authors clarified that Seg2Any is modular, with each component addressing a distinct failure mode, and provided quantitative evidence for each. They reimplemented PLACE fairly under identical FLUX settings, showing Seg2Any’s superiority (+2.9% MIoU ADE20K, +4.1% SACap-Eval), which resolved concerns about ADE20K performance. They explained that annotation noise in SACap-1M has limited effect, supported by qualitative evidence. They also promised to include missing attention visualizations and additional ablations in the camera-ready. Reviewers q31Y and T7Yi explicitly raised their scores after rebuttal, finding concerns addressed. Reviewer tBGK acknowledged that responses were thorough but lacking visualizations due to policy restrictions, maintained a borderline score. Reviewer Q4gs remained critical, emphasizing weaker global metrics (FID, IR, CLIP, PICK) and expressing skepticism despite clarifications.

Recommendation:
Overall, three reviewers (q31Y, T7Yi, tBGK) are inclined to accept, noting novelty, strong results, and valuable dataset contributions. One reviewer (Q4gs) remains unconvinced due to concerns about generative quality metrics. Given the technical innovations, extensive evaluation, clear community contribution of a dataset and benchmark, and convincing rebuttal addressing most substantive issues, I recommend accept.